# MOORL: A Framework for Integrating Offline-Online Reinforcement Learning

**Gaurav Chaudhary**                                              *gauravch@iitk.ac.in*
*Department of Electrical Engineering*
*Indian Institute of Technology Kanpur*

**Washim Uddin Mondal**                                           *wmondal@iitk.ac.in*
*Department of Electrical Engineering*
*Indian Institute of Technology Kanpur*

**Laxmidhar Behera**                                              *lbehera@iitk.ac.in*
*Department of Electrical Engineering*
*Indian Institute of Technology Kanpur*

**Reviewed on OpenReview:** *https://openreview.net/forum?id=PHsfZnF2FC*

## Abstract

Sample efficiency and exploration remain critical challenges in Deep Reinforcement Learning (DRL), particularly in complex domains. Offline RL, which enables agents to learn optimal policies from static, pre-collected datasets, has emerged as a promising alternative. However, offline RL is constrained by issues such as out-of-distribution (OOD) actions that limit policy performance and generalization. To overcome these limitations, we propose Meta Offline-Online Reinforcement Learning (MOORL), a hybrid framework that unifies offline and online RL for efficient and scalable learning. While previous hybrid methods rely on extensive design components and added computational complexity to utilize offline data effectively, MOORL introduces a meta-policy that seamlessly adapts across offline and online trajectories. This enables the agent to leverage offline data for robust initialization while utilizing online interactions to drive efficient exploration. Our theoretical analysis demonstrates that the hybrid approach enhances exploration by effectively combining the complementary strengths of offline and online data. Furthermore, we demonstrate that MOORL learns a stable Q-function without added complexity. Extensive experiments on 28 tasks from the D4RL and V-D4RL benchmarks validate its effectiveness, showing consistent improvements over state-of-the-art offline and hybrid RL baselines. With minimal computational overhead, MOORL achieves strong performance, underscoring its potential for practical applications in real-world scenarios.

## 1 Introduction

Deep Reinforcement Learning (DRL) has been tremendously successful in solving a variety of complex problems, including robotics (Tang et al., 2025), autonomous driving (Kiran et al., 2021), healthcare (Yu et al., 2021), game-playing (Silver et al., 2017; Vinyals et al., 2019), intelligent perception system (Chaudhary et al., 2023), and finance (Charpentier et al., 2021). However, one of the primary drawbacks of DRL algorithms is their sample inefficiency, i.e., the number of state-action-state transition samples they require to train a policy. Typically, these algorithms require millions of such interactions, making them impractical in real-world scenarios, particularly in safety-critical domains like robotics and autonomous driving (Kiran et al., 2021). Learning policies in controlled simulation environments can offer a partial remedy, as these policies often fail to generalize to real-world situations due to the well-known simulation-to-reality (sim2real) gap (Tobin et al., 2017).

An effective strategy to mitigate these problems is Offline Reinforcement Learning (RL) (Levine et al., 2020), which enables policy learning from historical datasets without requiring online exploration. Offline RL can train policies safely and cost-effectively using pre-collected human demonstrations or previously logged interactions. Although it alleviates some concerns related to sample complexity, the reliance on static offline data introduces new challenges, such as extrapolation errors and out-of-distribution (OOD) actions (Bai et al., 2022), which can result in sub-optimal behaviors when policies are tested in real environments or unfamiliar states (Kim et al., 2024). Offline-to-Online (O2O) RL (Lee et al., 2022; Wagenmaker & Pacchiano, 2023) dilutes the limitations of purely offline RL to some extent. O2O RL setups typically pre-train the agent using offline data, followed by fine-tuning via limited online interactions. However, this approach often experiences performance drops due to compounded Bellman errors (Sun et al., 2023)due to changes in reward distributions and distributional shifts between offline data and online interaction (Farahmand et al., 2010; Munos, 2005).

In this article, we aim to tackle the inherent challenges faced by offline RL and online RL algorithms. We believe that directly integrating offline data into online RL training can lead to more stable learning and mitigate issues such as out-of-distribution (OOD) actions and inefficient exploration. While recent efforts, such as RLPD (Ball et al., 2023) and Hy-Q (Song et al., 2022), have attempted to address offline-online integration, each comes with its own challenges. RLPD, which combines offline and online data, introduces extensive design components (refer to Appendix E). RLPD employs a large Q-ensemble and a high Update-to-Data (UTD) ratio to stabilize learning and optimize performance. These dependencies add complexity to the algorithm, making it more challenging and computationally intensive, limiting its scalability.

On the other hand, Hy-Q offers a more streamlined approach by integrating offline data into online training without necessitating extensive design components. However, Hy-Q has its own limitations; it requires maintaining separate Q-value functions for each timestep within a fixed horizon, significantly increasing computational and memory overhead, especially in environments with longer or variable horizons. Additionally, Hy-Q's reliance on a predefined horizon makes it less adaptable to tasks with dynamic episode lengths, limiting its scalability and flexibility across diverse RL scenarios. While Hy-Q reduces the need for many design components compared to RLPD, it still struggles with computational efficiency and adaptability in more complex or heterogeneous environments.

Further, as highlighted by (Furuta et al., 2021; Engstrom et al., 2019; Henderson et al., 2018), the RL algorithms are difficult to optimize and tune, where minor hyperparameter changes can have a non-trivial impact on performance. We believe it is important to limit RL algorithm design components. These limitations underscore the necessity for a hybrid approach that effectively integrates offline and online data and maintains computational efficiency and adaptability across various tasks. With this motivation in this work, we propose a framework called Meta Offline-Online RL (MOORL) that addresses the stated issues by utilizing a unified set of design components without the need for a large Q-ensemble, high UTD, and separate Q-function per horizon step, offering a more robust and generalizable solution for hybrid reinforcement learning. In particular, our contributions can be summarized as follows.

- We provide theoretical insights showing that mixing offline and online data can affect performance and provide a performance bound on the expected reward.

- We leverage the off-policy RL framework, Soft-Actor-Critic (SAC) (Haarnoja et al., 2018), to seamlessly integrate offline and online data via meta-learning for efficient design-free policy learning without introducing any new hyperparameters.

- Our proposed framework, MOORL, uses meta-learning principles (Finn et al., 2017; Nichol & Schulman, 2018) to train policies under a single meta-objective, enabling the dynamic balancing of offline and online data. The learned meta-policy adapts across varying distributions, minimizing the impact of distributional shifts and extrapolation errors.

- We validate our methodology through 28 comprehensive experiments on benchmark D4RL (Fu et al., 2020) and V-D4RL (Lu et al., 2022) environments, demonstrating that MOORL outperforms state-of-the-art methods in reward accumulation while being stable across diverse tasks, including dense and sparse reward scenarios.

## 2 Preliminaries

### 2.1 Markov Decision Process

We consider a Markov Decision Process (MDP) $\mathcal{M} = \langle \mathcal{S}, \mathcal{A}, T, H, R, \gamma, \rho \rangle$, where $\mathcal{S}$ indicates the state space, $\mathcal{A}$ denotes the action space, $T : \mathcal{S} \times \mathcal{A} \to \Delta(\mathcal{S})$ represents the state transition dynamics (where $\Delta(\mathcal{S})$ defines the collection of all probability distributions over $\mathcal{S}$), $R : \mathcal{S} \times \mathcal{A} \times \mathcal{S} \to \mathbb{R}$ is the reward function, $\gamma \in (0, 1)$ is the discount factor, $H$ is the length of the horizon of the episodes, and $\rho \in \Delta(\mathcal{S})$ is the initial state distribution. At each time instant $t$, the agent observes the state $s_t$, executes an action $a_t$, and as a result, transitions to the next state $s_{t+1} \sim T(\cdot|s_t, a_t)$, and receives a reward $r_t = R(s_t, a_t, s_{t+1})$. The goal in reinforcement learning is to learn a (stochastic stationary) policy $\pi : \mathcal{S} \to \Delta(\mathcal{A})$ that maximizes the expected cumulative reward $J_\pi = \mathbb{E}\left[\sum_{t=0}^{H-1} \gamma^t r_t \mid \pi, \rho\right]$ where the expectation is obtained over $\pi$-induced trajectories of length $H$ that start from the initial state distribution, $\rho$. The state-action distribution under policy $\pi$ is defined as $d^\pi(s, a) = (1 - \gamma) \sum_{t=0}^{\infty} \gamma^t P(s_t = s, a_t = a \mid \pi)$. The state-value and state-action value functions are defined, respectively, as $V^\pi(s) = \mathbb{E}\left[\sum_{t=0}^{H-1} \gamma^t r_t \mid s_0 = s, \pi\right]$ and $Q^\pi(s, a) = \mathbb{E}\left[\sum_{t=0}^{H-1} \gamma^t r_t \mid s_0 = s, a_0 = a, \pi\right]$, where the expectations are obtained over $\pi$-induced trajectories of length $H$. Note that $J_\pi = \mathbb{E}_{s \sim \rho}[V^\pi(s)]$. In this paper, we obtain the optimal policy that maximizes $J_\pi$ by combining offline and online data using ideas from meta-reinforcement learning, which is described below.

### 2.2 Meta-Reinforcement Learning

Meta-RL aims to solve a distribution of tasks given by $\mathcal{P}_\mathcal{T}(\cdot)$, rather than a single fixed task, where each task is characterized by an MDP $\mathcal{M}_i = \langle \mathcal{S}_i, \mathcal{A}_i, T_i, H_i, R_i, \gamma_i, \rho_i \rangle$. In our setting, the meta-learning process is used to combine offline and online data. Following the meta-RL paradigm, we maintain separate replay buffers for each type of data, denoted as $\mathcal{D}_{\text{offline}}$ and $\mathcal{D}_{\text{online}}$ respectively, which store transition tuples in the form $(s_i, a_i, r_i, s_i')$. The training process alternates between sampling from these replay buffers to update the policy while adapting to the changing data as the agent gathers more online experiences. A meta-episode consists of sampling data from either of these replay buffers and forming the associated trajectories to update the meta policy. A significant challenge arises from the distributional shift between offline and online data, which can lead to instability in the training process. To mitigate this issue, our meta-RL approach employs gradient-based meta-learning to effectively balance updates between the offline and online data sources, enhancing the robustness of the policy against distribution mismatches.

## 3 Why Meta Learning?

Meta-learning, or "learning to learn," is a powerful paradigm that performs well across diverse task distributions (Finn et al., 2017; Nichol & Schulman, 2018). By leveraging meta-learning to integrate offline and online data, Meta Offline Reinforcement Learning (MOORL) can achieve the following benefits:

- **Reduced Extrapolation Error and Improved Distribution Generalization:** MOORL adapts to changing data distributions by training a meta-policy over online and offline data. This approach minimizes extrapolation errors by optimizing the meta-policy to generalize across distributions rather than being confined to a single dataset (Finn et al., 2017; Garcia & Thomas, 2019).

- **Improved Credit Assignment:** MOORL utilizes a meta-objective that optimizes task performance across various data sources, helping isolate beneficial behaviors. By employing gradient-based meta-learning techniques, the meta-policy can assign credit more accurately, focusing on the trajectory aspects that generalize well between online and offline data (Finn et al., 2017; Al-Shedivat et al., 2018).

- **Multi objectivity:** MOORL streamlines the learning process by employing a single meta-objective that integrates agent and expert data without extensive design choices. This meta-objective is robust across varying distributions, automating many design decisions that would require manual adjustments (Chen et al., 2019; Ye et al., 2021).

## 4 Integrating Online RL with Offline Data

This work aims to bridge the gap between online and offline reinforcement learning by presenting a unified approach that integrates both paradigms without introducing extensive design elements and large computational overhead. Our method utilizes an off-policy reinforcement learning algorithm (Haarnoja et al., 2018) to effectively leverage data from various distributions. The proposed framework, MOORL, is designed to handle variations in offline data quality, making it adaptable to various scenarios. Additionally, MOORL demonstrates consistent performance across different problem settings, including environments with state-based or pixel-based observations, as well as those with dense, sparse, or even binary rewards. To support this, we first offer theoretical insights into why combining offline data with online agent data may be a more effective strategy than relying solely on online learning. We then introduce the MOORL framework, highlighting its simplicity and computational efficiency (refer to Appendix E, D).

### 4.1 Expected Reward and its Performance Bound

In this section, we examine the expected cumulative reward when sampling trajectories from a mixed distribution $\mathcal{D}$, which combines data generated by an offline policy $\mu$ and an online policy $\pi$. We derive a performance bound for the expected reward difference relative to the online policy, leveraging the total variation distance as a measure of distributional discrepancy between the offline and online state-action visitation distributions.

#### 4.1.1 Problem Setup

Consider an offline policy $\mu$, which generates the offline dataset, and an online policy $\pi$, which interacts with the environment. Let $\mathcal{D}$ denote a mixed dataset of trajectories, where a fraction $\lambda \in [0,1]$ of the trajectories are sampled from $\mu$, and the remaining $1 - \lambda$ are sampled from $\pi$. The state-action visitation distribution induced by a trajectory randomly sampled from $\mathcal{D}$ is defined as:

$$d^{\mathcal{D}}(s,a) = \lambda d^{\mu}(s,a) + (1 - \lambda)d^{\pi}(s,a) \tag{1}$$

Here, $d^{\mu}(s,a)$ and $d^{\pi}(s,a)$ represent the state-action visitation distributions under policies $\mu$ and $\pi$, respectively, over the state-action space $\mathcal{S} \times \mathcal{A}$. The expected cumulative reward generated by trajectories sampled from $\mathcal{D}$ is given by (Durugkar, 2023):

$$\mathbb{E}_{\mathcal{D}}[R] = \frac{1}{1 - \gamma} \sum_{(s,a)} d^{\mathcal{D}}(s,a)R(s,a) \tag{2}$$

Here, $R(s,a) = \mathbb{E}_{s' \sim T(\cdot|s,a)}[R(s,a,s')]$.

Our goal is to quantify the performance difference between the expected reward under the mixed distribution $\mathcal{D}$ and that under the online policy $\pi$:

$$\Delta R = \mathbb{E}_{\mathcal{D}}[R] - \mathbb{E}_{\pi}[R] \tag{3}$$

#### 4.1.2 Performance Gain Expression

The expected reward under the online policy $\pi$ is:

$$\mathbb{E}_{\pi}[R] = \frac{1}{1 - \gamma} \sum_{(s,a)} d^{\pi}(s,a)R(s,a) \tag{4}$$

Thus, the performance gain $\Delta R$ becomes:

$$\Delta R = \frac{1}{1 - \gamma} \sum_{(s,a)} \left( d^{\mathcal{D}}(s,a) - d^{\pi}(s,a) \right) R(s,a) \tag{5}$$

Substituting $d^{\mathcal{D}}(s,a) = \lambda d^{\mu}(s,a) + (1-\lambda)d^{\pi}(s,a)$, we obtain:

$$\Delta R = \frac{\lambda}{1-\gamma} \sum_{(s,a)} \left(d^{\mu}(s,a) - d^{\pi}(s,a)\right) R(s,a) \tag{6}$$

This expression indicates that the performance gain depends on the difference between the offline and online distributions, weighted by $R$ scaled by $\lambda$.

### 4.1.3 Bounding the Performance Gain with Total Variation Distance

To bound $\Delta R$, we introduce the Total Variation (Cover, 1999) distance between the distributions $d^{\pi}(s,a)$ and $d^{\mu}(s,a)$. The Total Variation distance between $d^{\pi}(s,a)$ and $d^{\mu}(s,a)$ is defined as:

$$\text{TV}(d^{\pi}, d^{\mu}) = \frac{1}{2} \sum_{(s,a)} |d^{\pi}(s,a) - d^{\mu}(s,a)| \tag{7}$$

The total variation distance quantifies the maximum difference in probability assigned to any event by the two distributions and satisfies $0 \leq \text{TV}(d^{\pi}, d^{\mu}) \leq 1$. It provides a natural metric for comparing $d^{\pi}(s,a)$ and $d^{\mu}(s,a)$ in the context of performance analysis.

For any bounded function $f(s,a)$:

$$\left| \sum_{(s,a)} (d^{\mu}(s,a) - d^{\pi}(s,a)) f(s,a) \right| \leq \sum_{(s,a)} |(d^{\mu}(s,a) - d^{\pi}(s,a))| \cdot |f|_{\infty} \tag{8}$$

where $|f|_{\infty} = \max_{(s,a)} R(s,a)$. If the reward function is taken to be bounded, i.e., $|R(s,a)| \leq R_{\max}$, $\forall (s,a)$, then from equation 8 and the definition of TV (equation 7), we arrive at the following.

$$\left| \sum_{(s,a)} (d^{\mu}(s,a) - d^{\pi}(s,a)) R(s,a) \right| \leq 2 \cdot \text{TV}(d^{\pi}, d^{\mu}) \cdot R_{\max} \tag{9}$$

This leads to the following bound on the performance gain.

$$|\Delta R| \leq TV(d^{\pi}, d^{\mu}) \cdot \frac{2\lambda R_{\max}}{1-\gamma} \tag{10}$$

Pinsker's inequality (Pinsker, 1964) provides an upper bound on the TV distance between two probability distributions in terms of their Kullback-Leibler (KL) (Kullback & Leibler, 1951)divergence:

$$\text{TV}(d^{\pi}, d^{\mu}) \leq \sqrt{\frac{1}{2} \mathbb{D}_{\text{KL}}(d^{\pi} \parallel d^{\mu})} \tag{11}$$

Using this definition, the final performance bound can be defined as:

$$|\Delta R| \leq \sqrt{2\mathbb{D}_{\text{KL}}(d^{\pi} \parallel d^{\mu})} \cdot \frac{\lambda R_{\max}}{1-\gamma} \tag{12}$$

The above result suggests that the performance difference between the mixed distribution $\mathcal{D}$ and the online policy $\pi$ is influenced by the total variation distance $\text{TV}(d^{\pi}, d^{\mu})$ and the mixing parameter $\lambda$. Specifically, when $\text{TV}(d^{\pi}, d^{\mu})$ is small, that the offline and online policies induce similar state-action distributions, the expected reward under $\mathcal{D}$ closely approximates that under $\pi$, suggesting stability in integrating the two data sources. Conversely, a large $\text{TV}(d^{\pi}, d^{\mu})$ implies a potentially significant deviation in cumulative rewards, highlighting the risk of distributional mismatch when combining offline and online data.

Importantly, the performance gain $\Delta R$ depends on the relative quality of the offline policy $\mu$ compared to the online policy $\pi$. For instance, if $\mu$ is an expert policy, $\mathbb{E}_{\mathcal{D}}[R] > \mathbb{E}_{\pi}[R]$ is expected early in training when $\pi$ is suboptimal, resulting in a positive $\Delta R > 0$. However, as training progresses and $\pi$ improves, $\text{TV}(d^{\pi}, d^{\mu})$ may decrease, reducing the benefit of mixing. This dynamic suggests that the value of offline data changes over time, with the greatest benefits often realized in the initial training phase.

Crucially, this analysis is not limited to expert offline datasets. For non-expert datasets, where $\mu$ may be suboptimal or diverse, the evolving difference between $d^{\pi}$ and $d^{\mu}$ still governs the effectiveness of data mixing. In such cases, an adaptive approach is essential to manage the shifting relationship between offline and online distributions, ensuring that the policy can leverage useful information from offline data while avoiding negative impacts as $\pi$ matures. This motivates the design of MOORL, which employs meta-learning to dynamically balance offline and online data, optimizing performance across diverse dataset qualities and training stages.

## 4.2 MOORL: Meta Offline-Online RL

This section introduces the Meta Offline-Online Reinforcement Learning (MOORL) framework, which addresses the challenges of integrating offline and online data in off-policy reinforcement learning. MOORL combines the strengths of off-policy learning (Haarnoja et al., 2018) and meta-learning (Finn et al., 2017), leveraging offline data for efficient learning while enabling online exploration, all while ensuring stable Q-learning updates. By employing a meta-learning strategy to dynamically adapt Q-function updates, MOORL mitigates issues such as distributional mismatch, overestimation bias, and instability in Q-learning.

### 4.2.1 Problem Definition

The task is to learn a policy using two distinct data distributions:

- **Offline Data** ($\mathcal{D}_{\text{offline}}$): This data consists of trajectories previously collected from one or more policies. While offline data may include high-reward sequences, it is often derived from sub-optimal or outdated policies, leading to potential biases. Direct incorporation of this data into training may cause overestimation bias in learned Q-values due to limited diversity and representativeness of experiences.

- **Online Data** ($\mathcal{D}_{\text{online}}$): This data is collected through interactions with the environment based on the current policy. Initially, online data may yield low rewards due to early-stage exploration, but typically improves as the policy refines.

The primary challenge comes from the distributional mismatch between offline and online data. Directly mixing these two distributions in off-policy RL algorithms can lead to *instability in Q-learning*, as the value estimates can become biased towards the high-reward offline data, resulting in overestimated Q-values. The proposed MOORL framework addresses this by learning a meta Q-function $Q_{\text{meta}}(s, a; \theta_{\text{meta}})$, parameterized by $\theta_{\text{meta}}$, that aims to generalize across both offline and online data distributions. The meta Q-function is optimized using a meta-Q objective using Reptile (Nichol & Schulman, 2018) meta-learning algorithm, balancing contributions from offline and online data.

### 4.2.2 Meta Q-Function Learning

MOORL learns robust meta Q-values that generalize across offline and online data distributions to mitigate overestimation bias. The MOORL framework aims to learn a meta Q-function $Q(s, a; \theta_{\text{meta}})$, parameterized by $\theta_{\text{meta}}$, that generalizes across both offline and online distributions. The parameter $\theta_{\text{meta}}$ can be interpreted as a solution to the Bellman error minimization problem stated below.

$$\min_{\theta_{\text{meta}}} \mathbb{E}_{(s,a,r,s')\sim\mathcal{D}} \left[ \left( Q(s,a;\theta_{\text{meta}}) - \left( r + \gamma \mathbb{E}_{s'\sim T(s'|s,a)} \left[ \max_{a'} Q(s',a';\theta_{\text{meta}}) \right] \right) \right)^2 \right], \tag{13}$$

---

**Algorithm 1** MOORL: Meta Offline-Online Reinforcement Learning

---

1: **Initialize:** Meta-policy parameters actor $\phi_{\text{meta}}$, critic $\theta_{\text{meta}}$, and temperature $\alpha$.
2: Offline dataset $\mathcal{D}_{\text{offline}}$ and empty online buffer $\mathcal{D}_{\text{online}}$.
3: Meta-learning rate $\eta_{\text{meta}}$, inner-loop learning rate $\eta$, number of iterations $N$.
4: **for** $n = 1$ to $N$ **do**
5:     **Select Buffer:** Choose $\mathcal{D}_{\text{offline}}$ or $\mathcal{D}_{\text{online}}$ as the data buffer.
6:     **Inner-loop Adaptation:**
7:     Collect transition in online environment and store in $\mathcal{D}_{\text{online}}$.
8:     Sample mini-batch from the selected data buffer.
9:     Perform $K$ inner actor $\tilde{\phi}$ and critic $\tilde{\theta}$ updates using data from $\mathcal{D}_i$.
10:     **Meta-update:**
11:     Update meta-policy parameters of both actor and critic using $\phi_{\text{meta}} \leftarrow \phi_{\text{meta}} - \eta_{\text{meta}}\nabla_{\phi_{\text{meta}}}[\mathcal{L}(\tilde{\phi})]$ and
    $\theta_{\text{meta}} \leftarrow \theta_{\text{meta}} - \eta_{\text{meta}}\nabla_{\theta_{\text{meta}}}[\mathcal{L}(\tilde{\theta})]$, respectively.
12: **end for**

---

where $\mathcal{D}$ is a dataset containing both offline and online data.

The combined loss ensures that $Q(\cdot, \cdot; \theta_{\text{meta}})$ provides consistent value estimates across both offline and online data. However, this objective is similar to mixing offline and online data distribution as done by RLPD (Ball et al., 2023), which simply mixes data from different distributions and learns a Q-function that can generalize across data distributions, which requires different design choices to stabilize learning. In this work, we apply a meta-learning perspective. Specifically, our algorithm progresses in multiple epochs, performing one meta-policy update at each epoch. At the start of an epoch, we randomly choose either the offline or the online replay buffer and perform $K$ inner updates for distribution adaptation, followed by a meta-update.

Specifically, for a given data distribution, $i \in \{\text{offline}, \text{online}\}$, we take a mini-batch $\mathcal{B}_i \subset \mathcal{D}_i$ of length $B$, and define the following loss functions for inner loop adaptation of the sampled distribution.

$$\mathcal{L}_i^{\text{critic}}(\theta) = \mathbb{E}_{(s,a,r,s') \sim \mathcal{B}_i}\left[\left(Q(s,a;\theta) - (r + \gamma Q(s',a';\theta'))\right)^2\right] \tag{14}$$

where $\theta'$ is the parameter of a target function that is used for stabilizing the learning. Following $Q$-learning paradigm, $\theta$ and $\theta'$ are synced periodically. The above loss is used to update the critic parameter, $\theta$, via a gradient-descent approach. The weights of the critic in the inner update loop are initialized with meta-critic weights, followed by $K$ inner loop gradient steps for distribution adaptation. where $\eta$ denotes a learning parameter. Mathematically, each gradient descent step is defined as follows.

$$\theta \leftarrow \theta - \eta\nabla_\theta\mathcal{L}_i^{\text{critic}}(\theta) \tag{15}$$

Let the final critic parameter (after $K$ inner loop steps) be $\tilde{\theta}$. We similarly define an actor loss function as follows, where $\phi$ denotes the parameter of the actor (policy approximator), and $\alpha$ is a tunable parameter.

$$\mathcal{L}_i^{\text{actor}}(\phi, \theta) = \mathbb{E}_{s \sim \mathcal{B}_i, a \sim \pi(\cdot|s,\phi)}\left[\alpha \log \pi(a|s; \phi) - Q(s,a;\theta)\right] \tag{16}$$

Starting from meta-policy weights, the parameter $\phi$ is also updated $K$ times as follows.

$$\phi \leftarrow \phi - \eta\nabla_\phi\mathcal{L}_i^{\text{actor}}(\phi, \theta) \tag{17}$$

Here, we use the principle of Soft Actor Critic (Haarnoja et al., 2018) that maximizes the expected reward (Q-value) while ensuring sufficient exploration through entropy regularization controlled by the temperature parameter $\alpha$. The entropy term encourages the policy to remain stochastic, promoting diverse actions and balancing exploration and exploitation. Let the final value of the actor parameter be $\tilde{\phi}$. After inner loop distribution adaptation, in the outer loop, we update the meta actor and critic parameters $\theta_{\text{meta}}, \phi_{\text{meta}}$ using the updated inner loop parameters as follows, where $\eta_{\text{meta}}$ is a tunable learning parameter.

$$\theta_{\text{meta}} \leftarrow \theta_{\text{meta}} - \eta_{\text{meta}}\nabla_{\theta_{\text{meta}}}[\mathcal{L}(\tilde{\theta})] \tag{18}$$

$$\phi_{\text{meta}} \leftarrow \phi_{\text{meta}} - \eta_{\text{meta}}\nabla_{\phi_{\text{meta}}}[\mathcal{L}(\tilde{\phi}, \tilde{\theta})] \tag{19}$$

For computing meta-updates $\phi_{\text{meta}}$ and $\theta_{\text{meta}}$ in the last step in the direction of $\phi_{\text{meta}} - \tilde{\phi}$ and $\theta_{\text{meta}} - \tilde{\theta}$, we treat $\phi_{\text{meta}} - \tilde{\phi}$ and $\theta_{\text{meta}} - \tilde{\theta}$ as a gradient similar to (Nichol & Schulman, 2018) and plunge it into an adaptive algorithm such as Adam (Kingma, 2014). In summary, the learning process consists of two steps: first, adapting the distribution through $K$ inner updates, followed by a meta-update aimed at generalizing across distributions

Hence, by leveraging meta-learning capabilities of task adaptation (Finn et al., 2017), we enable adaptation across diverse data distributions generated by different policies. This distribution adaptation strategy allows the learned Q-function to approximate a combined Bellman Q-function. Further, Figure 1 illustrates that the Q-values learned via MOORL and RLPD exhibit similar convergence trends. This demonstrates that learning a combined Bellman function, as done by RLPD or exploring a meta Q-function, yields comparable Q-values. However, MOORL offers the advantage of not relying on specific design choices for stability, highlighting its robustness and simplicity.

## 5 Experiments

We evaluate the proposed MOORL approach on the D4RL benchmark (Fu et al., 2020) and V-D4RL (Lu et al., 2022), comparing its performance against state-of-the-art methods, including hybrid RL approach RLPD (Ball et al., 2023), Hy-Q (Song et al., 2022) and offline RL method ReBRAC (Tarasov et al., 2024), for completeness. Each baseline has distinct design choices and operational paradigms that provide valuable context for evaluating MOORL's strengths in hybrid offline-online RL settings. Through our chosen baselines and benchmark tasks, we aim to address the following questions:

- Can MOORL effectively integrate offline data into an online RL setting without extensive design-specific configurations?

- Does MOORL ensure stable learning across diverse data distributions, minimizing the need for task-specific tuning?

- Can MOORL be extended to high-dimensional image-based observation data?

- Does MOORL maintain consistent performance across varying offline data qualities by learning a stable Q-function?

### 5.1 Evaluation on Offline D4RL Tasks

To evaluate MOORL's performance, we select a range of D4RL tasks to assess robustness across diverse data distributions:

- **D4RL Locomotion**(Fu et al., 2020): This set includes 15 dense-reward locomotion tasks, with offline data covering varying levels of optimality, from expert to random trajectories.

- **D4RL Maze-Navigation**(Fu et al., 2020): We utilize 6 AntMaze navigation tasks with sparse binary reward structures, each with different complexities.

- **D4RL Adroit** (Fu et al., 2020): The tasks in this set (Pen, Door, Hammer) involve complex manipulation and sparse rewards, with offline data consisting of expert-level trajectories.

#### 5.1.1 Choice of Baselines

**Hybrid RL:** We evaluate the performance of MOORL against current state-of-the-art hybrid RL approaches including RLPD (Ball et al., 2023) and Hy-Q (Song et al., 2022). These approaches aim to integrate offline and online learning. To ensure stable learning, they introduce many design elements, resulting in computational overhead and limiting their broader impact on real-world scenarios.

Table 1: The performance comparison across Locomotion tasks is presented as the average normalized score across 10 seeds. The symbol $\pm$ denotes the standard error of the mean.

| Task Name | TD3+BC | ReBRAC | Hy-Q | RLPD (UTD=20) | MOORL, our |
|---|---|---|---|---|---|
| half-cheetah-expert | $93.4 \pm 0.1$ | $105.9 \pm 0.6$ | $84.0$ | $\mathbf{111.1 \pm 0.1}$ | $\underline{105.9 \pm 0.3}$ |
| half-cheetah-medium-expert | $89.1 \pm 1.9$ | $101.1 \pm 1.7$ | $86.0$ | $\mathbf{105.8 \pm 0.1}$ | $\underline{103.4 \pm 1.0}$ |
| half-cheetah-medium-replay | $45.0 \pm 0.4$ | $51.0 \pm 0.3$ | $\underline{89.0}$ | $72.2 \pm 0.1$ | $\mathbf{96.9 \pm 0.7}$ |
| half-cheetah-medium | $54.7 \pm 0.3$ | $65.6 \pm 0.3$ | $\underline{88.0}$ | $85.4 \pm 0.1$ | $\mathbf{99.2 \pm 0.7}$ |
| half-cheetah-random | $30.9 \pm 0.1$ | $29.5 \pm 0.5$ | $80.0$ | $\underline{85.1 \pm 3.3}$ | $\mathbf{99.0 \pm 1.6}$ |
| hopper-expert | $\underline{109.6 \pm 1.2}$ | $100.1 \pm 2.8$ | $54.0$ | $101.7 \pm 5.5$ | $\mathbf{111.6 \pm 1.4}$ |
| hopper-medium-expert | $87.8 \pm 3.5$ | $\mathbf{107.0 \pm 2.1}$ | $100.0$ | $97.1 \pm 4.2$ | $\underline{101.5 \pm 2.0}$ |
| hopper-medium-replay | $55.1 \pm 10.6$ | $\underline{98.1 \pm 1.8}$ | $77.0$ | $81.3 \pm 0.8$ | $\mathbf{99.6 \pm 1.4}$ |
| hopper-medium | $60.9 \pm 2.5$ | $102.0 \pm 0.3$ | $\underline{106.0}$ | $90.8 \pm 0.7$ | $\mathbf{107.9 \pm 0.8}$ |
| hopper-random | $8.5 \pm 0.2$ | $8.1 \pm 0.8$ | $80.0$ | $\underline{92.9 \pm 0.9}$ | $\mathbf{101.9 \pm 1.1}$ |
| walker2d-expert | $110.0 \pm 0.2$ | $112.3 \pm 0.1$ | $112.0$ | $\mathbf{127.5 \pm 1.9}$ | $\underline{123.6 \pm 1.2}$ |
| walker2d-medium-expert | $110.4 \pm 0.2$ | $111.6 \pm 0.1$ | $95.0$ | $\mathbf{128.0 \pm 1.8}$ | $\underline{117.2 \pm 2.5}$ |
| walker2d-medium-replay | $68.0 \pm 6.4$ | $77.3 \pm 2.6$ | $103.0$ | $\underline{105.7 \pm 2.4}$ | $\mathbf{111.2 \pm 1.2}$ |
| walker2d-medium | $77.7 \pm 1.0$ | $82.5 \pm 1.2$ | $86.0$ | $\mathbf{115.1 \pm 2.1}$ | $\underline{114.1 \pm 0.6}$ |
| walker2d-random | $2.0 \pm 1.2$ | $18.4 \pm 1.5$ | $\underline{90.0}$ | $73.7 \pm 10.6$ | $\mathbf{93.8 \pm 0.3}$ |
| **Average** | $66.9 \pm 3.0$ | $78.0 \pm 1.0$ | $88.7$ | $\underline{98.2 \pm 3.7}$ | $\mathbf{105.8 \pm 1.1}$ |

Table 2: The performance comparison across AntMaze tasks is reported as the average normalized score over 10 seeds. The symbol $\pm$ represents the standard error of the mean. For RLPD, results with UTD=20 and UTD=5 correspond to 300K timesteps, whereas UTD=1 results are based on 1M timesteps.

| Task Name | TD3+BC | ReBRAC | Hy-Q | RLPD | | | MOORL, our |
|---|---|---|---|---|---|---|---|
| | | | | UTD=20 | UTD=5 | UTD=1 | |
| antmaze-umaze | $66.3 \pm 2.1$ | $97.8 \pm 0.3$ | - | $\underline{99.0 \pm 0.2}$ | $99.0 \pm 0.2$ | $88.2 \pm 9.3$ | $\mathbf{99.2 \pm 0.2}$ |
| antmaze-umaze-diverse | $53.8 \pm 2.8$ | $88.3 \pm 4.3$ | - | $97.8 \pm 0.3$ | $\underline{98.5 \pm 0.3}$ | $93.4 \pm 1.2$ | $\mathbf{99.0 \pm 0.3}$ |
| antmaze-medium-play | $26.5 \pm 6.1$ | $84.0 \pm 1.4$ | $25.0$ | $\mathbf{98.5 \pm 0.2}$ | $97.5 \pm 0.5$ | $94.4 \pm 1.2$ | $\underline{98.2 \pm 0.3}$ |
| antmaze-medium-diverse | $25.9 \pm 5.1$ | $76.3 \pm 4.5$ | $2.0$ | $\underline{98.0 \pm 0.3}$ | $97.5 \pm 0.3$ | $93.6 \pm 1.4$ | $\mathbf{98.5 \pm 0.4}$ |
| antmaze-large-play | $0.0 \pm 0.0$ | $60.4 \pm 8.7$ | $0.0$ | $\mathbf{88.0 \pm 0.8}$ | $75.0 \pm 8.3$ | $57.8 \pm 5.4$ | $\underline{82.3 \pm 3.4}$ |
| antmaze-large-diverse | $0.0 \pm 0.0$ | $54.4 \pm 8.4$ | $0.0$ | $\mathbf{87.5 \pm 1.2}$ | $77.5 \pm 4.2$ | $50.2 \pm 6.9$ | $\underline{85.6 \pm 2.1}$ |
| **Average** | $28.7 \pm 2.7$ | $76.8 \pm 4.6$ | $6.8$ | $\mathbf{94.8 \pm 0.5}$ | $90.8 \pm 2.3$ | $79.6 \pm 3.5$ | $\underline{93.8 \pm 1.1}$ |

**Offline RL:** For the offline RL method, we use ReBRAC (Tarasov et al., 2024), which builds on offline RL and investigates the effect of many design elements on the performance highlighting how different design elements can enhance the performance of offline RL agents. ReBRAC is primarily designed for offline RL, while it is less relevant to MOORL's hybrid offline-online framework, and is included for completeness.

**RL+BC:** These approaches use a minimalistic approach for learning from offline data. To demonstrate the effectiveness of MOORL against the Behavior Cloning (BC) regularized (Bain & Sammut, 1995) RL method, we use TD3+BC (Fujimoto & Gu, 2021) and DrQ+BC (Yarats et al., 2021) for state and pixel-based tasks.

### 5.1.2 Empirical Results

The baselines TD3+BC and ReBRAC results are taken from (Tarasov et al., 2024) while Hy-Q and DrQ+BC results are taken from (Nakamoto et al., 2024). Our results demonstrate that MOORL achieves competitive performance with RLPD, Hy-Q, and ReBRAC across the D4RL and V-D4RL tasks while being simple and computationally efficient without introducing many design components. MOORL exhibits stable learning in a hybrid RL setting, i.e., offline-online learning, and achieves robust cumulative rewards.

Table 3: The performance comparison across Adroit tasks is presented as the average normalized score across 10 seeds. The symbol $\pm$ denotes the standard error of the mean.

| Task Name | BC | TD3+BC | ReBRAC | RLPD (UTD=20) | MOORL, our |
|---|---|---|---|---|---|
| pen | 85.1 | $146.3 \pm 2.4$ | $\textbf{154.1} \pm \textbf{1.8}$ | $137.8 \pm 2.4$ | $\underline{150.0 \pm 0.8}$ |
| door | 34.9 | $84.6 \pm 14.8$ | $104.6 \pm 0.8$ | $\underline{105.5 \pm 2.1}$ | $\textbf{107.1} \pm \textbf{0.7}$ |
| hammer | 125.6 | $117.0 \pm 10.3$ | $133.8 \pm 0.2$ | $\textbf{140.3} \pm \textbf{2.8}$ | $\underline{137.2 \pm 1.1}$ |
| **Average** | 81.9 | $116 \pm 9.2$ | $\underline{130.8 \pm 1.0}$ | $127.9 \pm 2.4$ | $\textbf{131.4} \pm \textbf{0.9}$ |

On D4RL locomotion benchmarks, MOORL achieves the highest average performance. For the tasks with suboptimal data, hybrid approaches such as MOORL, Hy-Q, and RLPD highlight the advantage of using a hybrid learning approach. It is evident from Table 1 that MOORL performs most optimally across tasks, specifically where offline data is suboptimal. MOORL achieves $8 - 10\%$ performance improvement over RLPD without using large critic ensembles, high UTD, and layer normalization. Further, MOORL highlights that for the task, such as *halfcheetah-random*, it achieves a significant performance gain. We believe that this advantage comes from MOORL's adaptation strategy. Unlike other hybrid approaches, MOORL alternates between offline and online data sources rather than directly mixing them. This design prevents the policy from being overly constrained by the offline dataset at each adaptation step, as the inner-loop fine-tuning is performed separately for each data source. Consequently, MOORL shifts its reliance toward online interactions, resulting in superior policy.

The performance of RLPD is competitive to MOORL on the AntMaze navigation task, but as highlighted in Table 2, the performance of RLPD drops significantly with a lower UTD ratio (UTD=1), whereas MOORL performs significantly superior. The performance of RLPD (UTD=5), with a similar number of gradient steps, is similar to MOORL. However, MOORL being larger Q-ensemble free highlights its computational efficiency. For the D4RL AntMaze navigation tasks, MOORL avoids incorporating design choices such as Clipped Double Q-learning (CDQ) (Fujimoto et al., 2018) and Entropy Backup similar to RLPD, which are used in other tasks. These design choices tend to perform poorly in the sparse reward structure of AntMaze tasks, leading to overly conservative learning and suboptimal policy performance. While RLPD demonstrates the best performance under standard configurations, achieving a modest $3\%-4\%$ improvement over MOORL, its reliance on a high Update-to-Data (UTD) ratio makes it sensitive to this hyperparameter. With UTD=1, RLPD suffers a substantial performance drop, whereas MOORL maintains stable learning, achieving a $13\%-17\%$ improvement over RLPD at 1M timesteps. This highlights MOORL's ability to maintain effective learning without reliance on aggressive update schedules, making it particularly advantageous in scenarios with limited computational budgets or where low UTD ratios are preferred.

Adroit tasks pose significant challenges due to their high-dimensional action spaces and sparse reward structures. To evaluate the performance of various approaches, we conducted experiments on Adroit tasks using high-quality offline expert data. As shown in Table 3, the performance of all methods, including MOORL, remains consistent across tasks, with no single approach demonstrating a definitive advantage. Depending on the specific task, RLPD, ReBRAC, and MOORL occasionally outperform one another. However, the differences in performance are not statistically significant. This lack of clear improvement can likely be attributed to the inherent complexity of the Adroit tasks, which makes it challenging for any single method to achieve a distinct edge over others in this domain.

## 5.2 MOORL's Adaptability to Pixel-Based Observation Spaces

This section investigates MOORL's ability to operate effectively with high-dimensional image-based observations. Unlike methods that require extensive hyperparameter tuning or specialized architectural adjustments, MOORL seamlessly extends to this challenging domain by utilizing a shared feature extraction encoder within its actor-critic framework. This encoder processes raw pixel input into meaningful feature representations, enabling efficient learning even in visually complex environments. We evaluate MOORL's performance on

Table 4: The performance comparison across V-D4RL Locomotion tasks is presented as the average normalized score across 10 seeds. The symbol $\pm$ denotes the standard error of the mean.

| Task Name | BC | DrQ+BC | ReBRAC | RLPD | MOORL, our |
|-----------|-----|--------|--------|------|-----------|
| walker-walk-expert | $91.5 \pm 1.3$ | $68.4 \pm 2.5$ | $81.4 \pm 3.3$ | $\underline{91.5 \pm 1.1}$ | $\mathbf{94.6 \pm 0.2}$ |
| walker-walk-medium | $40.9 \pm 1.3$ | $46.8 \pm 0.8$ | $52.5 \pm 1.1$ | $\underline{84.9 \pm 0.7}$ | $\mathbf{87.9 \pm 1.8}$ |
| cheetah-run-expert | $\underline{67.4 \pm 2.3}$ | $34.5 \pm 2.8$ | $35.6 \pm 1.8$ | $\mathbf{68.3 \pm 1.0}$ | $53.2 \pm 2.5$ |
| cheetah-run-medium | $51.6 \pm 0.5$ | $\underline{53.0 \pm 1.0}$ | $\mathbf{59.0 \pm 0.2}$ | $49.0 \pm 1.3$ | $49.9 \pm 1.6$ |
| Average | $62.9 \pm 1.3$ | $50.7 \pm 1.8$ | $57.1 \pm 1.6$ | $\mathbf{73.4 \pm 1.0}$ | $\underline{71.4 \pm 0.9}$ |

high-dimensional DeepMind Control Suite (DMC) (Tassa et al., 2018) tasks, leveraging datasets with pixel-based observations that test robustness and adaptability. As shown in Table 4, MOORL outperforms RLPD on 3 out of 4 DMC tasks, while RLPD achieves slightly better average cumulative performance. Though RLPD remains competitive, its reliance on large Q-ensembles makes it computationally expensive, particularly for image-based tasks. MOORL, on the other hand, achieves comparable results without using large critic-ensembles, emphasizing its practicality for resource-constrained scenarios.

These findings highlight MOORL's capacity for generalization and computational efficiency in high-dimensional reinforcement learning tasks. By maintaining strong performance without complex design alterations or fine-tuning, MOORL demonstrates its suitability for visually demanding environments, reaffirming its flexibility and reliability across diverse settings.

### 5.3 Does MOORL learn Stable Q-Values

To this end, we demonstrate that MOORL enables stable learning without relying on extensive design choices. Specifically, we present the mean Q-values in Figure 1 to showcase the stability of MOORL's learning architecture. Despite the data quality, MOORL consistently learns a stable meta Q-function, translating into a stable meta-policy.

In Figure 1, we evaluate MOORL stability in the challenging and sparsely rewarded AntMaze navigation task. The diverse dataset is created by assigning random goal locations in the maze and directing the agent to navigate to them. In contrast, the play dataset consists of trajectories from specific, hand-picked initial positions to hand-picked goal locations. Remarkably, MOORL achieves similar Q-value trajectories across these datasets, emphasizing its robustness to variations in data quality.

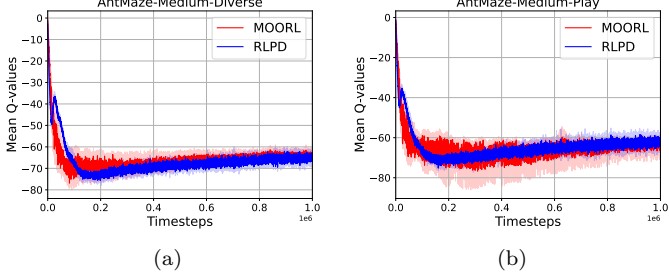

Figure 1: Learning curves showing the mean Q-values for the AntMaze-Medium task on Diverse and Play datasets. Figures 1a and 1b depict the performance of the MOORL and RLPD frameworks, respectively, demonstrating their learning stability and effectiveness across both datasets.

For comparison, we conducted a similar analysis on RLPD to further substantiate the stability of Q-value learning across different frameworks. This analysis highlights MOORL's capability to maintain stable learning dynamics even under diverse and challenging data conditions without requiring specific design choices.

## 6 Related Work

### 6.1 Offline Reinforcement Learning

Offline reinforcement learning (RL) has gained significant attention for its ability to learn policies from pre-collected datasets without additional environmental interaction. However, distributional shift remains a

fundamental challenge, as highlighted by Levine et al. (2020), designing algorithms that generalize effectively from limited data is crucial. Several approaches address this challenge by incorporating regularization techniques. TD3+BC (Fujimoto & Gu, 2021) integrates behavioral cloning into the actor loss to constrain policy learning, while CQL (Kumar et al., 2020) enforces conservatism by penalizing the critic for assigning high values to out-of-distribution (OOD) actions. IQL (Kostrikov et al., 2021) takes a different approach by leveraging advantage-weighted regression to avoid sampling OOD actions altogether. More recent methods further improve policy learning by incorporating representation learning, such as pre-training action encoders (Akimov et al., 2022; Chen et al., 2022) or estimating dataset uncertainty using VAEs (Wu et al., 2022) and RND (Nikulin et al., 2023). Another line of work improves policy robustness by leveraging ensemble-based uncertainty estimation. SAC-N (An et al., 2021) achieves strong results using large Q-function ensembles, though some tasks require ensembles of up to 500 members, making it computationally expensive. MSG (Ghasemipour et al., 2022) mitigates this by introducing independent target updates, reducing the ensemble size to four in MuJoCo tasks but still requiring 64 members for AntMaze.

Further, Fujimoto & Gu (2021) shows that non-algorithmic factors significantly influence performance, and ReBRAC (Tarasov et al., 2024) performs a detailed analysis to understand the effect of design choice on the performance of offline RL. Offline RL also struggles when datasets lack full coverage, making pessimism computationally challenging (Jin et al., 2021; Zhang et al., 2022) and often requiring strong representation conditions (Xie et al., 2021). To overcome these limitations, hybrid approaches incorporating limited online interaction have been proposed as a promising alternative.

## 6.2  Bridging Online and Offline RL

Integrating online and offline reinforcement learning (RL) is a critical research area. Empirical studies have examined how online learners can leverage logged data to improve performance (Rajeswaran et al., 2017; Nair et al., 2017; Hester et al., 2017; Ball et al., 2023; Nakamoto et al., 2024; Zheng et al., 2023). While practical benefits are evident, formal guarantees in this setting remain limited. (Ross & Bagnell, 2012) proposed a framework where a learner can choose between executing a logging policy $\mu$ or an alternative online policy, effectively bridging the gap between online and offline data exploitation. (Xie et al., 2021) demonstrated that no approach could achieve strictly better sample complexity than purely online or offline methods when using data collected from a logging policy. This result highlights the challenges of balancing online exploration with offline data utilization. In contrast, our research assumes access to a pre-collected offline dataset and the ability to interact online, enabling the refinement of a near-optimal policy while minimizing online interactions. (Song et al., 2022) proposed "Hybrid RL" using the Hybrid Q-learning algorithm (Hy-Q) for low bilinear rank Markov Decision Processes (MDPs) (Du et al., 2021). Under certain conditions, Hy-Q can achieve optimal policies efficiently. However, the method's performance may degrade when offline data coverage of the optimal policy is insufficient, illustrating the importance of comprehensive offline data. Further, a study by (Xie et al., 2022) delves into purely online contexts or frameworks involving offline datasets, offering insights into the concealability coefficient—a parameter critical in establishing guarantees for offline RL. This approach bridges the analytical methods used in online and offline settings. Recent work by (O'Donoghue et al., 2018; Wagenmaker & Pacchiano, 2022), and others explore synergies between these methodologies, revealing strategies for effectively integrating offline data with online exploration.

Our approach builds on these foundations by addressing sample complexity when merging offline datasets with online learning. We aim to enhance the unified integration of offline data into online reinforcement learning without extensive design choices and added computational complexity (Ball et al., 2023; Song et al., 2022), which limits the border applicability of such hybrid-RL approaches.

## 7  Conclusion

In this work, we demonstrated that off-policy RL can effectively integrate offline and online data distributions. By learning a meta-policy over these distributions, our approach enables stable Q-value estimation independent of data quality. Extensive experiments across 28 diverse tasks, spanning both state and pixel observations, validated the efficacy of our method. Specifically, we showed that a hybrid RL policy improves sample efficiency and ensures robust performance across varying data qualities. By limiting design compo-

nents and computational overhead, our approach generalizes well across different environments, making it a scalable and practical solution for real-world applications. These findings highlight the potential of combining offline and online learning to address key challenges in reinforcement learning. While MOORL demonstrates robustness in handling diverse offline datasets, highly biased data may still constrain exploration. Future work could explore adaptive mechanisms to mitigate such biases by dynamically adjusting the influence of offline data to further enhance learning stability and generalization.

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

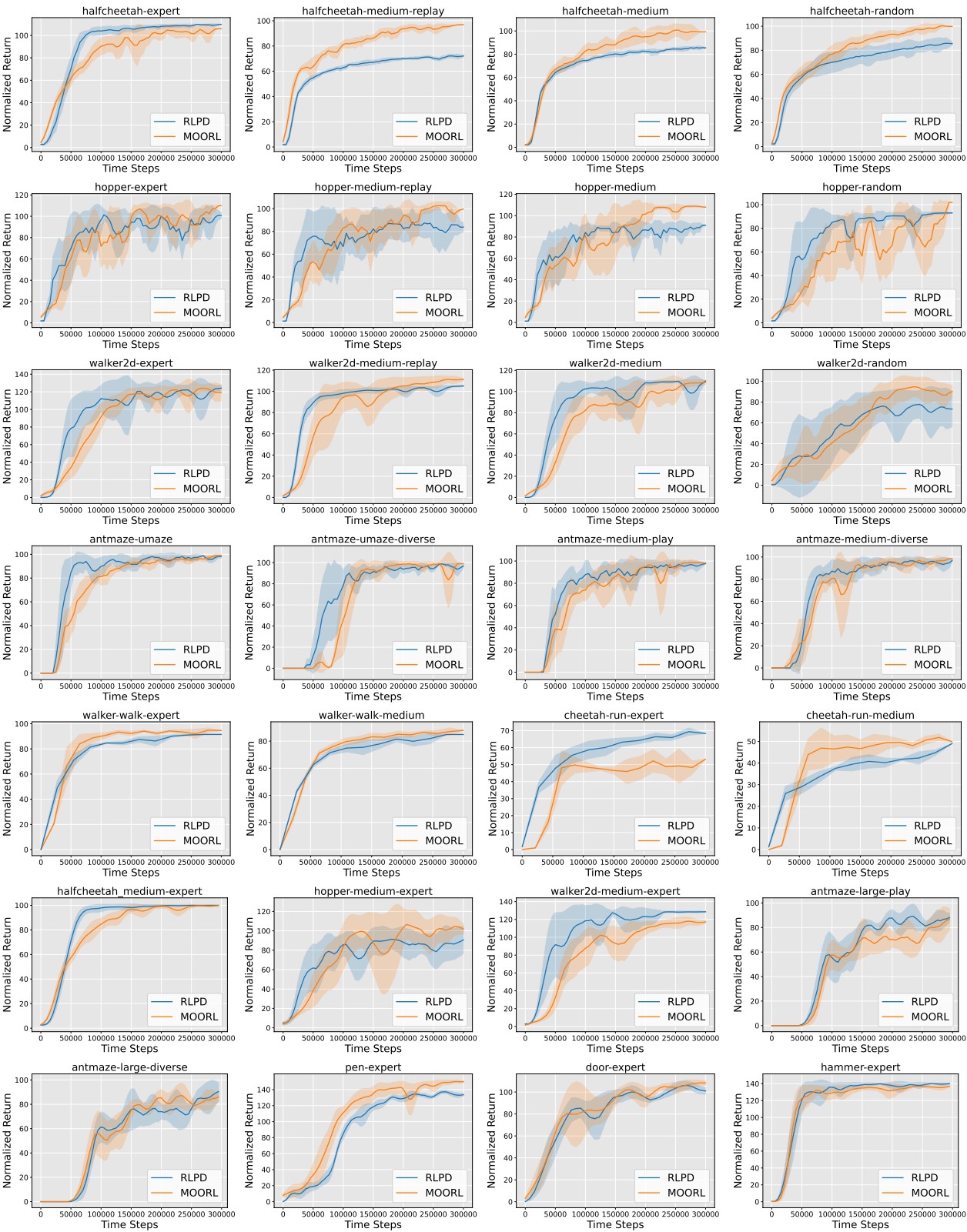

Figure 2: The plots show learning curves with normalized returns on the y-axis. Each curve represents the mean performance across 10 random seeds, with shaded regions indicating the standard deviation. The normalized return at each point is computed as the average over 10 evaluation episodes. All tasks are evaluated over 300K timesteps.

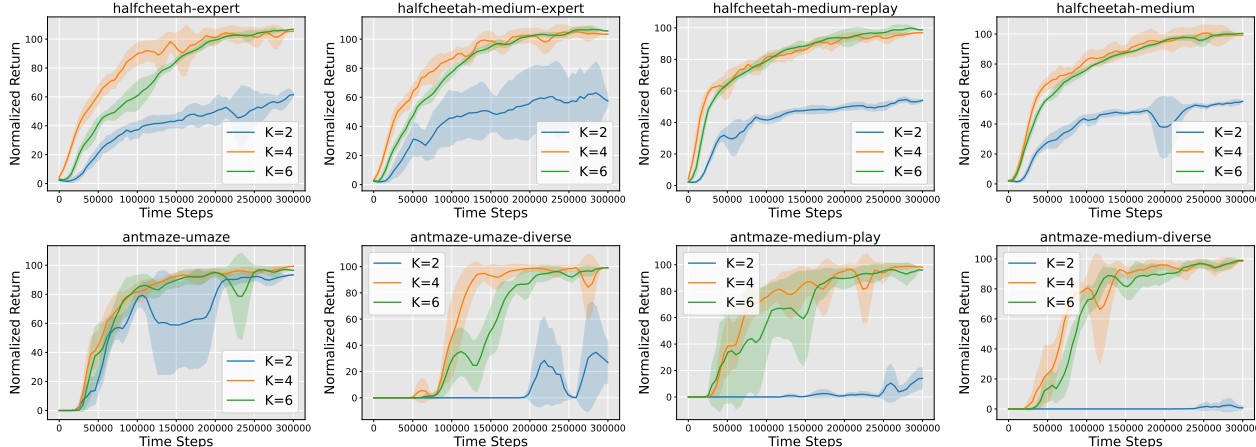

Figure 3: The plots illustrate the impact of the inner-loop adaptation step on learning. The y-axis represents the normalized return, while the x-axis denotes timesteps. The solid curves show the mean return across 10 random seeds, with shaded regions indicating the standard deviation. Each evaluation point is computed as the average return over 10 episodes. All tasks are evaluated over 300K timesteps.

# A  Ablation Study: Impact of Inner-Loop Adaptation Steps

The number of inner-loop adaptation steps ($K$) plays a crucial role in our meta-learning framework, influencing the policy's ability to adapt to distributional shifts. To assess its impact, we conduct an ablation study (Figure 3) by varying $K \in \{2, 4, 6\}$ while keeping all other hyperparameters fixed.

Our results highlight the importance of selecting an appropriate $K$ for effective adaptation. With $K = 2$, the model struggles to align with the target distribution, leading to high variance and degraded performance. This suggests that too few updates hinder the agent's ability to capture distributional shifts, resulting in unstable learning dynamics. Conversely, $K = 4$ provides the best balance between stability and adaptability, yielding optimal performance. However, increasing $K$ further to $K = 6$ offers no significant performance gains. Instead, we observe diminishing returns, where additional updates increase computational overhead without substantial improvements. This is likely due to overfitting to recent experiences, reducing the learned policy's generalization capability.

This aligns with prior meta-learning findings (Finn et al., 2017; Nichol & Schulman, 2018), which indicate that a moderate number of inner-loop steps maximizes generalization while maintaining efficient adaptation. Overall, our findings suggest that a balanced adaptation strategy—rather than aggressive inner-loop optimization—is key to achieving strong generalization in hybrid offline-online reinforcement learning.

# B  Implementation Details

To implement the proposed framework, we use entropy-regularized SAC (Haarnoja et al., 2018) as the base RL algorithm and apply Reptile (Nichol & Schulman, 2018) for meta-updates, improving generalization across distributions. The policy and Q-networks are 2-layer MLPs with 256 hidden units and ReLU activations. We use the Adam optimizer (Kingma, 2014) for inner-loop adaptation and meta-updates. Each training iteration consists of $K = 4$ SAC updates followed by a Reptile-style (Nichol & Schulman, 2018) meta-update. A learning rate of $3 \times 10^{-4}$ for inner-loop adaptation is used, while for meta-updates, the learning rate is dynamically adjusted (Nichol & Schulman, 2018) based on the ratio of the current timestep to

Table 5: MOORL Hyperparameters

| Parameter | Value |
|-----------|-------|
| Batch size | 256 |
| Discount ($\gamma$) | 0.99 |
| Optimizer | Adam |
| Learning rate | $3 \times 10^{-4}$ |
| Critic EMA Weight ($\rho$) | 0.005 |
| Inner Gradient Steps ($K$) | 4 |
| Network Width | 256 Units |
| Number of Layers | 2 |
| Initial Entropy Temperature ($\alpha$) | 1.0 |
| Target Entropy | $-\frac{\dim(A)}{2}$ |

total timesteps. We maintain an exponentially moving average target Q-network with an update weight of $\rho = 0.005$. Hyperparameters are summarized in Table 5.

## C  Inner-Loop Adaptation in MOORL vs. UTD in RLPD

It is essential to distinguish the role of inner-loop adaptation steps ($K$) from the Update-To-Data (UTD) ratio in RLPD (Ball et al., 2023). The UTD ratio dictates the number of gradient updates per environment step, primarily influencing sample efficiency in off-policy RL by controlling the degree of data reuse. In contrast, our adaptation steps ($K$) govern the number of inner-loop within each outer-loop meta-update iteration, directly shaping the adaptation dynamics rather than the frequency of gradient updates. While a high UTD enables more updates per collected transition, $K$ determines how effectively the learned policy aligns with the target distribution across offline and online phases. Thus, $K$ plays a fundamental role in optimizing meta-adaptation rather than modulating sample efficiency, making it a distinct and crucial design choice in our framework. These distinctions become evident in Figure 1, where we show that both RLPD and MOORL learn similar Q values. While MOORL inherently adapts to changing distribution through inner and meta updates, RLPD uses layer norm and a large Q-ensemble to stabilize learning, essentially to learn similar Q-values.

## D  Computational Cost Comparison

We compare the computational cost of our methods with RLPD on Adroit hand tasks, including pen door and hammer. The RLPD for these tasks uses a 3-layer MLP, whereas MOORL uses a 2-layer MLP, which is consistent across all the tasks as highlighted in Table 7. Also, RLPD uses a high UTD ratio (20) while maintaining a large Q-ensemble (10), which is in contrast to MOORL, which maintains a simple Double-Q network architecture. RLPD is trained for 300K timesteps at each timestep, it performs 20 gradient steps given by UTD, essentially resulting in 6M gradient steps over 300K timesteps. MOORL performs 4 inner update adaptation gradient steps and 1 meta update gradient step at each timestep, resulting in a total of 1.5M gradient steps over 300K timesteps. RLPD takes approx 0.5sec while MOORL takes 0.05sec per timestep when run on a single RTX A4000 GPU.

## E  MOORL: Embracing Simplicity in Design

Recent works have demonstrated that leveraging offline data can significantly enhance reinforcement learning performance. State-of-the-art methods in hybrid RL (e.g., RLPD (Ball et al., 2023)) and offline RL (e.g.,

ReBRAC (Tarasov et al., 2024)) typically rely on extensive component selection and fine-tuning of design elements to address offline data challenges. In contrast, MOORL adopts a simpler algorithmic structure that requires fewer design components.

MOORL integrates offline and online data using an architecture that minimizes reliance on extensive tuning—avoiding the need for deep network ensembles or high UTD ratios—while still effectively handling distribution shifts and limited exploration. As illustrated in Tables 6 and 7, MOORL delivers robust performance with a streamlined set of design choices compared to ReBRAC and RLPD.

Although the MOORL algorithm is not entirely free of design decisions, the simplicity of MOORL's approach enhances its adaptability across diverse tasks, providing a more generalizable and robust solution relative to methods that depend heavily on intricate design configurations.

Table 6: Comparison of Design Choices with ReBRAC (Tarasov et al., 2024)

| Component | ReBRAC | MOORL |
|---|---|---|
| Deeper Networks | ✓ | ✗ |
| Larger Batches | ✓ | ✗ |
| Layer Normalization | ✓ | ✗ |
| Decoupled Penalization | ✓ | ✗ |
| Adjusted Discount Factor | ✓ | ✗ |

Table 7: Comparison of Design Choices with RLPD (Ball et al., 2023)

| Component | RLPD | MOORL |
|---|---|---|
| Sampling Strategy | ✓ | ✗ |
| Layer Normalization | ✓ | ✗ |
| Random Ensemble Distillation | ✓ | ✗ |
| Clipped Double Q-Learning | ✓ | ✓ |
| Network Architectures | ✓ | ✗ |
| Update to Data Ratio | ✓ | ✗ |
| Entropy Backup | ✓ | ✓ |

# F   Detailed Task Definition

## F.1   D4RL: Locomotion

In these tasks, the reward is dense and based on the agent's forward velocity, penalizing large control inputs to encourage stable movement. The goal is to maximize the cumulative reward over the episode by learning an efficient and stable gait. The standard evaluation metric is the normalized score, which is computed by normalizing the agent's return relative to expert and random policies, as defined by (Fujimoto et al., 2019). The datasets are generated from policies of varying expertise, including *random*, *medium*, *medium-replay*, *medium-expert*, and *expert* trajectories. Episodes typically last for 1,000 timesteps without early termination.

## F.2   D4RL: AntMaze

In these tasks, the reward is sparse and binary, indicating whether the agent has reached the goal. Upon reaching the goal, the episode terminates. The normalized return is measured as the proportion of successful trials out of evaluation trials following prior work. The dataset consists of *play-based* and *diverse* demonstrations, where the former includes goal-directed trajectories, and the latter contains broader movement data. The challenge in this domain arises from long-horizon credit assignment and the need for effective exploration in a sparse reward setting.

### F.3  D4RL: Adroit

The Adroit suite consists of dexterous hand manipulation tasks that require controlling a 24-DoF simulated Shadow Hand robot to perform complex actions such as hammering a nail, opening a door, or twirling a pen. This domain is specifically designed to assess the impact of narrowly distributed expert demonstrations on learning in a high-dimensional robotic manipulation setting with sparse rewards. Unlike standard Gym MuJoCo tasks, Adroit exhibits several distinct characteristics. First, the dataset is sourced from human demonstrations. Second, solving these tasks with online RL alone proves challenging due to the sparse reward structure and inherent exploration difficulties. Lastly, the high-dimensional nature of these tasks introduces additional complexity in representation learning.

### F.4  V-D4RL: DeepMind Control Suite (DMC)

The DMC tasks involve controlling physics-based agents with dense rewards that encourage smooth, efficient movement. The standard evaluation metric is the normalized score, computed using the agent's return normalized against the performance of a well-trained SAC policy. The datasets include *expert* and *medium* policies, allowing evaluation of an agent's ability to learn from varying data quality. Episodes typically run for 1,000 timesteps without early termination.

