# OpenReview forum: "MOORL: A Framework for Integrating Offline-Online Reinforcement Learning"
_TMLR — Accepted by TMLR_

### Review · Reviewer_uLsg · 2025-02-20

**Summary Of Contributions:**

This paper presents an interesting perspective on viewing offline-online RL as a meta-learning problem. The proposed algorithm, MOORL, treats learning from the offline dataset and the online buffer as two separate meta-learning tasks and applies meta-learning to update the actor-critic. Benchmark experiments on 28 tasks demonstrate that MOORL can match or surpass state-of-the-art baselines.

**Audience:**

Yes

**Broader Impact Concerns:**

None.

**Claims And Evidence:**

No

**Requested Changes:**

- I request the authors to revise the paper thoroughly to ensure that the language and notation are rigorous and consistent.
- Please provide learning curves showing normalized returns for MOORL alongside hybrid baselines such as Hy-Q and RLPD. This will help demonstrate MOORL's sample efficiency and clarify how the reported numbers in the tables were obtained.
- The theoretical analysis of why the offline dataset benefits online RL should be revised. Specifically, the authors should clearly differentiate between $r(s, a)$, $Q^\pi(s, a)$, and $\log d^\pi(s, a)$ in the derivation and explicitly state any assumptions made.
- The claim that MOORL is "design-free" should be reconsidered or better justified. Both RLPD and MOORL involve design choices—RLPD focuses on the component level while MOORL focuses on the algorithmic level. The authors should clarify this distinction and avoid overstating the generality of MOORL.
- In Section 3, the authors list four benefits of incorporating meta-learning, but none of these claims are supported by literature or experiments. Please support these claims with references to prior work or empirical evidence through experiments.
- Please provide the ablation study on the inner-loop steps $K$. Additionally, make a fair comparison with RLPD on AntMaze if you intend to argue that its good performance is due to taking more gradient steps.
- Since RL is known to be sensitive to hyperparameters and random seeds, please provide sufficient implementation details, and ideally, also release the code.

**Strengths And Weaknesses:**

### Strengths
- The paper addresses a timely topic in RL and proposes a simple yet effective solution.
- The paper provides a comprehensive benchmark of MOORL on 28 tasks from four different suites.

### Weaknesses
- The paper is not well-written. Specifically, the language and notation are not rigorous and sometimes lead to confusion. Here are some examples:
    - In multiple instances, the authors refer to an "offline expert dataset," but the offline dataset can also be task-irrelevant, such as a random dataset.
    - In the captions of Tables 1–4, it is unclear what the authors mean by "unseen seeds."
    - On page 8, second paragraph, I assume you mean $\theta_{\mathrm{meta}} - \widetilde{\theta}$ since $\widetilde{\theta}_{\mathrm{meta}}$ is not defined anywhere.
    - In the pseudocode, line 7, are you sure you only update the actor-critic once after completing full trajectories of environment interactions? I believe most works update per environment step.
    - In the methods section, MOORL involves two learning rates: $\eta$ for the inner loop and $\eta_{\mathrm{meta}}$ for the outer loop. However, in the appendix, only one learning rate is listed.
- One of the core contributions of the paper—the theoretical analysis of why the offline dataset benefits online RL—appears problematic. I am not an expert in theory, but the authors seem to use $r(s, a)$, $Q^\pi(s, a)$, and $\log d^\pi(s, a)$ interchangeably throughout the derivation without any stated assumptions.
- One of the claims made by the authors is that MOORL is "design-free," which I find unconvincing. For example, both RLPD and MOORL explore the design of an offline-online algorithm, but from different perspectives: RLPD focuses on component selection, while MOORL focuses on algorithmic structure. In my opinion, neither of them is truly design-free. I agree that MOORL is a simple algorithm, but not that it is design-free.
- In Section 3, the authors list four benefits of incorporating meta-learning, but none of these claims are supported by references or experiments.
- The paper lacks ablation studies on the new hyperparameter, namely the inner-loop steps $K$. Since meta-learning effectively takes $K+1$ gradient steps to update the actor-critic once, $K$ plays a similar role to the Update-to-Data (UTD) ratio in RLPD. Given that $K$ is set to 4 in this paper, it would be more informative to compare it fairly to AntMaze with UTD=5 in RLPD.
- Although MOORL leverages an offline dataset, it remains an online RL method. However, the paper does not provide learning curves, which are crucial for evaluating the sample efficiency of an online RL method.

---

> ### Author Response · Authors · 2025-03-06
> **Response to Reviewer uLsg**
>
> We thank the reviewer for their valuable feedback and suggestions. We have incorporated their suggestion in the revised draft. Below, we address the question raised by the reviewer.
>
> **Requested changes and weakness 1: Writing suggestion**
>
> We have taken your feedback and incorporated all the writing suggestions.
>
> **Requested changes 2 and weakness 6: Learning curves**
>
> We have added learning curves highlighting normalized returns in the revised draft, which can be found in the Appendix Figure 2.
>
> **Reuested changes 3 and weakness 2: the authors should clearly differentiate between $r(s, a)$, $Q^\pi(s, a)$, and $\log d^\pi(s, a)$ in the derivation and explicitly state any assumptions made.**
>
> We politely disagree with the reviewer. We don't make any understated assumptions. All the notions are standard in RL settings, whose definitions can be found in the preliminary section with added definitions and clarity. In entropy-regularized reinforcement learning, such as Soft Actor-Critic (Haarnoja et al., 2018), the objective extends beyond immediate rewards to optimizing long-term performance. The action-value function,$
> Q^\pi(s,a) = \mathbb{E} [\sum_{t=0}^{\infty} \gamma^t r(s_t,a_t) \mid s_0=s, a_0=a ],$ captures the expected cumulative reward under policy $ \pi$. Entropy-regularized policies often follow a Boltzmann form, $ \pi(a\mid s) \propto \exp(Q^\pi(s,a)) $, linking Q-values to the induced state-action distribution,  $d^\pi(s,a) = (1 - \gamma) \sum_{t=0}^{\infty} \gamma^t P(s_t = s, a_t = a \mid \pi).$
> When evaluating a dataset $ \mathcal{D} $ of trajectories, the expected cumulative reward can be expressed as
>
> $\mathbb{E}_{\tau \sim \mathcal{D}}[R(\tau)] = \sum\_{(s,a)} d^\mathcal{D}(s,a) Q^\pi(s,a)$.
>
> This formulation highlights the interplay between rewards, Q-values, and state-action distributions, forming the basis for our analysis.
>
>
> **Reuested changes 4 and weakness 3: The claim that MOORL is "design-free" should be reconsidered or better justified. I agree that MOORL is a simple algorithm, but not that it is design-free.**
>
> We appreciate your concern about `MOORL' being design-free. Taking your feedback, we have toned down our claims about MOORL being design-free to avoid overstating that and also acknowledge this in the revised Appendix E. In the revised draft, we propose MOORL as simple and less computationally expensive, with supporting quantitative evidence in Appendix D and E.
>
> **Requested changes 6 and weakness 5: Please provide the ablation study on the inner-loop steps $K$ and comparison of MOORL with RLPD 5.**
>
> We have added an ablation study to highlight the effect of learning with varying numbers of inner loop updates for \{K=2,4,6\}. Through the
> ablation study, we found that for a small inner gradient step, the agent shows slow learning as a generalization
> to changing distribution becomes difficult, while for the higher values, it provides no added advantages. A
> more detailed discussion can be found in Appendix A.
>
>
> We have also included the results on Antmaze tasks for RLPD with UTD=5, which can be found in Table 2. Further, we would like to highlight that RLPD UTD=1 results were obtained by training for 1M timesteps.
>
> We understand that inner-loop adaptation steps (K) and the Update-To-Data (UTD) ratio in RLPD may seem similar, but there is a subtle difference. The UTD ratio controls data reuse and sample efficiency by dictating gradient updates per environment step, while K determines the number of inner-loop updates per meta-update, shaping adaptation dynamics. These distinctions become evident in Figure 1, where we show that both RLPD and MOORL learn similar Q values. While MOORL inherently adapts to changing distribution through inner and meta updates, RLPD uses layer norm and large Q-ensemble to stabilize learning, essentially to learn similar Q-values.
>
> **Requested changes 7:  please provide sufficient implementation details**
>
> We have added an implementation section to the appendix B and will release the code after acceptance.
>
>
> [1] Tuomas Haarnoja, Aurick Zhou, Kristian Hartikainen, George Tucker, Sehoon Ha, Jie Tan, Vikash Kumar,
> Henry Zhu, Abhishek Gupta, Pieter Abbeel, et al. Soft actor-critic algorithms and applications. arXiv
> preprint arXiv:1812.05905, 2018.

---

> > ### Author Response · Authors · 2025-03-06
> > **Regarding Section 3**
> >
> > **Requested changes 5 and weakness 4: In Section 3, the authors list four benefits of incorporating meta-learning, but none of these claims are supported by references or experiments.**
> >
> > We appreciate the reviewer’s feedback regarding Section 3. Our intent in that section was to provide intuition behind integrating meta-learning with offline and online data rather than to assert these benefits as novel contributions. Based on your suggestions, we have added appropriate citations to Section 3. We explain each point here and support each point with relevant citations.
> >
> >
> > - **Reduced Extrapolation Error and Improved Generalization:** Meta-learning frameworks like MAML (Finn et al., 2017) show that training across diverse task distributions enables policies to generalize with minimal adaptation. By meta-training on both offline and online data, MOORL mitigates extrapolation errors from distributional shifts (Garcia \& Thomas, 2019). Meta-policies leverage prior experience to adapt exploration strategies, reducing overfitting to a single dataset.
> >
> > - **Balanced Integration of Expert and Agent Behaviors:** RL with demonstrations (Rajeswaran et al., 2017; Nair et al., 2020) suggests that combining expert guidance with exploration improves robustness. MOORL extends this by dynamically interpolating between expert priors (offline data) and exploratory behaviors (online data), rather than merely imitating experts. This aligns with Nair et al. (2020), where policies trained with demonstrations exhibit robust behaviors, while meta-learning ensures balanced credit assignment.
> >
> > - **Improved Credit Assignment:** Gradient-based meta-learning optimizes policies for rapid adaptation by identifying transferable decision components (Finn et al., 2017). In MOORL, this isolates behaviors that generalize across offline and online data, improving credit assignment. Empirical results show MOORL outperforms RLPD and Hy-Q on the locomotion random dataset, validating its ability to discern high-reward actions in dynamic data distributions (Al-Shedivat et al., 2018).
> >
> > - **Simplified Algorithmic Complexity:** Meta-learning consolidates multi-objective optimization, reducing manual reward engineering (Chen et al., 2019). Instead of handcrafted reward weighting, meta-learning automates optimization, enhancing generalization across objectives (Ye et al., 2021). This aligns with MOORL’s approach, where meta-learning replaces manual data-balancing heuristics with a unified optimization process. Figure 1 illustrates that MOORL stabilizes Q-learning like RLPD but without requiring layer normalization or large Q-ensembles.
> >
> > [1] Chelsea Finn, Pieter Abbeel, and Sergey Levine. Model-agnostic meta-learning for fast adaptation of deep
> > networks. In International conference on machine learning, pp. 1126–1135. PMLR, 2017.
> >
> > [2] Francisco Garcia and Philip S Thomas. A meta-mdp approach to exploration for lifelong reinforcement
> > learning. Advances in Neural Information Processing Systems, 32, 2019.
> >
> > [3] Aravind Rajeswaran, Vikash Kumar, Abhishek Gupta, Giulia Vezzani, John Schulman, Emanuel Todorov,
> > and Sergey Levine. Learning complex dexterous manipulation with deep reinforcement learning and
> > demonstrations. RSS, 2017.
> >
> > [4] Ashvin Nair, Abhishek Gupta, Murtaza Dalal, and Sergey Levine. Awac: Accelerating online reinforcement
> > learning with offline datasets. arXiv preprint arXiv:2006.09359, 2020.
> >
> > [5] Maruan Al-Shedivat, Trapit Bansal, Yuri Burda, Ilya Sutskever, Igor Mordatch, and Pieter Abbeel. Contin-
> > uous adaptation via meta-learning in nonstationary and competitive environments. ICLR, 2018.
> >
> > [6] Xi Chen, Ali Ghadirzadeh, Mårten Björkman, and Patric Jensfelt. Meta-learning for multi-objective re-
> > inforcement learning. In 2019 IEEE/RSJ International Conference on Intelligent Robots and Systems
> > (IROS), pp. 977–983. IEEE, 2019.
> >
> > [7] Feiyang Ye, Baijiong Lin, Zhixiong Yue, Pengxin Guo, Qiao Xiao, and Yu Zhang. Multi-objective meta
> > learning. Advances in Neural Information Processing Systems, 34:21338–21351, 2021.

---

> ### Comment · Reviewer_uLsg · 2025-03-10
> **Response to the rebuttal**
>
> I appreciate the authors' rebuttal and the revision. But I would like to raise some further points:
>
> **Regarding the learning curve:** Thanks for providing this extra Figure 2 which helps to demonstrate the sample efficiency of MOORL. But there are only 20 tasks shown in the Figure 2 while the experiments are on 28 tasks. What about the other 8 tasks? And besides, in general, please do a more careful job when stitching the figures. Currently, the numbers in the first and second columns from Figure 2 and 3 are partially covered by the next column.
>
> **Regarding the theoretical analysis in Section 4.1.2:** Thanks for the clarification. Some of the confusion has been resolved, but there are still remaining issues:
> - Although the analysis is based on entropy regularized RL, for the paper to be self-contained, I would suggest the authors explicitly mention the relationship between $\log d^\pi(s,a)$ and $Q^\pi(s,a)$.
> - Based on your clarification, eq. 2 is wrong in the current version. The $r(s,a)$ should be replaced by $Q^\pi(s,a)$.
> - Could you further explain how you derive eq. 12 from eq. 7 and eq. 11 (similarly for deriving eq. 15 from eq. 7 and eq. 14)?
> - I don't get the point of eq. 8, eq. 9 and eq. 13. The definitions of $S_1$ ​and $S_2$ directly lead to eq. 10 and eq. 14 without the middle equations. Maybe I am missing something?
>
> **Regarding Section 3:** I appreciate the authors' attempt to make proper citations, but I still find some of them problematic:
> - For the second point, both of the citations are not related to Meta Learning. Generally speaking, I think this point holds true for any algorithm that works on this hybrid setting with both offline and online datasets. It is rather a benefit for the problem setting than the advantage of a family of algorithms. I would recommend removing this point.
> - For the fourth point, I think your new argument regarding the multi-objective tasks is much more convincing than the simplicity argument. Although I think the implementation of MOORL is simple within the scope of hybrid RL, meta learning itself with so many design choices is hardly considered simple. I would recommend replacing the fourth point with only the multi-objective argument.

---

> > ### Author Response · Authors · 2025-03-13
> > **Response to Reviewer uLsg**
> >
> > **Regarding Learning Curve:**
> >
> > We sincerely appreciate the reviewer’s attention to detail and thoughtful feedback. In Figure 2, we selectively visualized learning curves for a representative subset of tasks (20 out of 28) to ensure clarity and facilitate direct comparisons across domains. We emphasize that the omitted tasks (e.g., additional variants of locomotion or Adroit tasks) consistently align with the trends shown, and full results have been included in the updated draft.
> >
> > **Regarding Section 4.1.2**
> >
> > - We have corrected Equation 2.
> >
> > - Under the entropy regularized SAC framework (Haarnoja et al., 2018), the policy is defined as a Boltzmann distribution over Q-values, ensuring $ d^\pi(s, a) \propto \exp(Q^\pi(s, a))$ (Haarnoja et al., 2017).
> > As shown by (Levine, 2018), this entropy regularization introduces a KL divergence penalty into the policy objective. Combining this with the entropy-KL inequality (Equation 7), we derive Equation 11.
> >
> > - The revised draft has expanded the intermediate steps to reach Equation 12 from Equation 11. The same analysis can be extended to derive Equation 15 from Equation 14.
> >
> > - While Equations 10 and 14 could be derived directly from $S_1$ and $S_2$, Equations 8, 9, and 13 serve two critical roles:
> >     - Partitioning the State-Action Space: They rigorously define $S_1$ and $S_2$ based on visitation frequency ratios: $d^{\pi}(s,a) \geq d^{\mu}(s,a) \quad \text{vs.} \quad d^{\pi}(s,a) < d^{\mu}(s,a),$ ensuring the analysis is grounded in measurable distributional shifts.
> >
> >     - Connecting KL Divergence to Q-Values: Equation 8 explicitly shows how deviations between $d^{\pi}$ and $d^{\mu}$ penalize/amplify $Q^{\pi}(s,a)$ through the term: $ \exp \left(-\log \frac{d^{\pi}(s,a)}{d^{\mu}(s,a)} \right).$ This bridges the gap between distributional mismatch and performance bounds, making the theoretical insights more interpretable.
> >
> > These steps are essential for readers to follow why the KL divergence term acts as a penalty or bonus in different subsets of the state-action space. Removing them would obscure the connection between distributional shifts and policy performance.
> >
> > **Regarding Section 3**
> >
> > Taking your feedback, we have removed the second point and re-framed the fourth point for multi-objectivity.
> >
> >
> >
> > [1] Tuomas Haarnoja, Aurick Zhou, Kristian Hartikainen, George Tucker, Sehoon Ha, Jie Tan, Vikash Kumar, Henry Zhu, Abhishek Gupta, Pieter Abbeel, et al. Soft actor-critic algorithms and applications. arXiv preprint arXiv:1812.05905, 2018.
> >
> > [2] Haarnoja, Tuomas, et al. "Reinforcement learning with deep energy-based policies." International conference on machine learning. PMLR, 2017.
> >
> > [3] Levine, Sergey. "Reinforcement learning and control as probabilistic inference: Tutorial and review." arXiv preprint arXiv:1805.00909 (2018).

---

> ### Comment · Reviewer_uLsg · 2025-03-13
> **Problem remains regarding the equations**
>
> Thanks for your response. I still think there are problems with the equations:
> - Sorry for my typo in the previous response. I intended to ask about how you derive eq. 11 (not eq. 12) from eq. 7 and 10. The text you added still didn't clarify this.
> - Maybe eq. 8 shows some insight regarding policy learning, but for eq. 8, one can directly use the importance-sampling trick to have a much simpler term $\frac{d^\mu(s,a)}{d^\pi(s,a)}$ instead of the complicated $\exp(- \log \frac{d^\pi(s,a)}{d^\mu(s,a)})$.

---

> > ### Author Response · Authors · 2025-03-15
> > **Response to uLsg**
> >
> > 1. Here’s a concise explanation of how Equation 11 arises from Equations 7 and 10, focusing on the role of the KL divergence penalty:
> > - Equation 7 establishes a bound between the online policy’s state-action distribution ($d^\pi(s,a)$) and the offline data distribution ($d^\mu(s,a)$), penalized by their KL divergence. This ensures the online policy does not deviate arbitrarily from the offline data.
> >
> > - Equation 10 identifies regions ($S_1$) where the online policy dominates the offline data ($d^\pi(s,a) \geq d^\mu(s,a)$). Here, the offline data’s contribution to Q-values is inherently limited. However, this bound alone does not account for how much the online policy deviates from the offline data.
> >
> > - Equation 11 refines Equation 10 by introducing the KL divergence penalty from Equation 7. In $S_1$, the bound is tightened by subtracting this penalty, quantifying the mismatch between the online and offline distributions.
> >
> > - Why subtract the KL term?
> > The KL divergence is non-negative $(D_{\text{KL}}\geq0)$.  Subtracting it reduces the upper bound on the offline data’s contribution, making the inequality stricter. This ensures the policy pays a penalty proportional to its divergence from the offline data in $S_1$, preventing over-reliance on its own exploration.  Without subtraction, the bound would remain loose, failing to penalize over-exploration. Subtraction directly enforces the trade-off encoded in Equation 7.
> >
> > 2. We agree that expressing the ratio as $d^{\mu}(s,a) / d^{\pi}(s,a)$ simplifies the notation. However, we retained the exponential form in Equation (8) to align with the entropy-regularized framework (e.g., SAC). This choice emphasizes the connection to policy regularization and exploration dynamics inherent to our theoretical insights while ensuring consistency.

---

> > > ### Comment · Reviewer_uLsg · 2025-03-15
> > > **please provide a rigorous and detailed derivation**
> > >
> > > I must insist that you directly address my question. I am **not** asking about **why** Eq. 11 is derived, but rather **how** you derive it.
> > >
> > > To be explicit:
> > >
> > > - Eq. 10 does **not** directly lead to Eq. 11. The KL divergence term is non-negative, and subtracting it from the greater side does not guarantee that the inequality still holds.
> > > - Eq. 7 does not seem relevant here, as it concerns an inequality between  $\sum d^\pi(s, a) \log d^\pi(s, a)$ and $\sum d^\pi(s, a) \log d^\mu(s, a)$, whereas Eqs. 10 and 11 involve $\sum d^\pi(s, a) Q^\pi(s, a)$ and $\sum d^\mu(s, a) Q^\pi(s, a)$. Even assuming a Boltzmann distribution, i.e., $d(s, a) \propto \exp Q(s, a)$, the connection remains unclear.
> > >
> > > Your current response does not address this issue. If Eq. 11 is valid, please provide a **rigorous and detailed** derivation.

---

> > > > ### Author Response · Authors · 2025-03-20
> > > > **Response to uLsg**
> > > >
> > > > Dear Reviewer,
> > > >
> > > > We sincerely apologize for the lack of clarity in our earlier presentation of the theoretical concepts in Section 4.1.2. Your feedback has been invaluable in guiding us towards a more rigorous and effective explanation of our ideas.
> > > >
> > > > We have revised Section 4.1 to enhance its clarity and theoretical foundation in response to your suggestions. Specifically, we have introduced the total variation distance as a metric to quantify the distributional dissimilarity between the state-action visitation distributions of offline data and online data. This addition provides a precise framework for analyzing the performance implications of mixed data distribution, directly addressing the concerns you raised.
> > > >
> > > > In the revised section, we provide a concise theoretical analysis of the impact of the mixed data distribution $\mathcal{D}$ on the performance. We establish a performance bound for the expected difference in cumulative rewards between $\mathcal{D}$ and the online policy $\pi$. This bound, formulated as $|\Delta R|\leq \sqrt{2\mathbb{D}_{\text{KL}}(d^\pi\parallel d^\mu)} \cdot \frac{\lambda R\_{\max}}{1 - \gamma} $. This theoretical insight provides a clear understanding of how integrating offline and online data affects performance.
> > > >
> > > > It is important to note that while this theoretical discussion in Section 4.1 provides context, our primary contribution remains in the novel algorithmic design and robust experimental results demonstrating its efficacy. The theoretical analysis merely complements these findings.
> > > >
> > > > We trust that these revisions adequately address your concerns and provide a more compelling and coherent explanation. We appreciate your constructive feedback, which has significantly enhanced the quality of our paper. These changes reinforce the manuscript and align it more closely with the expected standards of clarity and rigor.

---

> > > > > ### Comment · Reviewer_uLsg · 2025-03-24
> > > > >
> > > > > Thanks for rewriting Section 4.1.
> > > > >
> > > > > First, in the new version, I think $R(s, a)$ is not clearly defined as it contradicts $r_t = R(s_t, a_t, s_{t+1})$ in Section 2.1. Please clarify the difference.
> > > > >
> > > > > Besides this, the new derivation seems mathematically correct to me. However, I am not sure what the point of this new analysis is. Based on the analysis, we only have a positive gain $\Delta R$ when the offline dataset is from an expert, while the majority of the experiments are done on non-expert datasets. The analysis in the end gives an upper-bound of the performance gain, which I don't find useful. Could you please clarify the motivation behind this analysis?
> > > > >
> > > > > Moreover, to be clear, I did not criticize the other contributions in this paper, which are acceptable to me. However, as you insist on framing the theoretical analysis as one of the main contributions of this paper (in the first point of Section 1), it needs to be correct and useful to the community.

---

> > > > > > ### Author Response · Authors · 2025-03-27
> > > > > >
> > > > > > Dear Reviewer,
> > > > > >
> > > > > > Thank you for your insightful and valuable feedback on our submission. Below, we clarify the purpose of the analysis and outline the revisions made to strengthen its presentation.
> > > > > > - **$R(s,a)$:** The definition $R(s_t,a_t,s_{t+1})$ and $R(s,a)$ seems to contradict but it is delibrate notational simplification. We have highlighted this in the revision.
> > > > > > -  **Relevance Across Dataset Types:** We apologize for lacking clarity here. The analysis, centered on the performance gain $\Delta R$, is not exclusive to expert data. It examines how $\Delta R$ depends on the difference between the offline distribution $d^\mu$  and the online distribution $d^\pi$, a factor relevant to any offline dataset. The sign of $\Delta R$ indicates whether offline data boosts or degrades performance based on the relative quality of $\mu$ and $\pi$. This insight is particularly valuable in non-expert settings, where adaptation becomes critical. To clarify this, we’ve added a paragraph in Section 4.1.3 emphasizing the analysis’s broad applicability and role in motivating an adaptive approach like MOORL.
> > > > > >
> > > > > > - **Usefulness of the Upper Bound:** We understand your concern that the upper bound on $|\Delta R|$ may not seem immediately practical. Our intent was not to predict the direction of performance changes but to quantify the potential magnitude of deviation driven by $TV(d^\pi,d^\mu)$. This bound serves as a diagnostic tool: a small distance suggests stability in mixing data, while a large distance signals the need for careful management of distributional shifts—a core challenge in hybrid RL. We’ve revised Section 4.1.3 to highlight this purpose, framing the bound as a motivator for adaptive strategies, which aligns with MOORL’s design.
> > > > > >
> > > > > > We believe these revisions address your concerns while enhancing the paper’s clarity and coherence. Thanks again for your thoughtful review, which has undoubtedly improved our work. We hope these changes meet your expectations.

---

> > > > > > > ### Comment · Reviewer_uLsg · 2025-03-27
> > > > > > >
> > > > > > > Thanks for the further clarification regarding the intention of the analysis. I think the new explanation does a much better job for positioning the analysis in the context of this paper, thus solves my concern.
> > > > > > >
> > > > > > > There is a small issue regarding the correctness: I don't think you can simply treat $R(s_t, a_t)$ as a simplified notation for $R(s_t, a_t, s_{t+1})$ since the transition $T(s_{t+1} | s_t, a_t)$ is stochastic. You probably need to define something like $R(s_t, a_t) = \int_{s_{t+1}} T(s_{t+1} | s_t, a_t) R(s_t, a_t, s_{t+1})$, or re-define the occupancy $d$ also on $(s_t, a_t, s_{t+1})$ for which I am not sure all the equations are still hold true.

---

> > > > > > > > ### Author Response · Authors · 2025-03-30
> > > > > > > >
> > > > > > > > We thank the reviewer for raising an important point regarding the notation of the reward function and its impact on the mathematical correctness of the reward bound. The difference between $ r_t = R(s_t, a_t, s_{t+1})$ in Section 2.1 and $ R(s, a) $ in Section 4.1 is not an inconsistency but a standard notational simplification in reinforcement learning. In Section 4.1, $ R(s, a) $ represents the *expected reward* over all possible next states, i.e., $ R(s, a) = \mathbb{E}_{s' \sim T(\cdot | s, a)} [R(s, a, s')] $. This expectation is essential to account for the stochasticity of the transition $ T(s' | s, a) $, ensuring that our analysis of expected returns and performance bounds is mathematically sound.
> > > > > > > >
> > > > > > > > We acknowledge the reviewer's concern that simply equating $ R(s, a) $ to $R(s, a, s') $ without considering the transition dynamics would be incorrect. However, by defining $R(s, a) $ as the expected reward, we properly incorporate the stochastic nature of the next state. This is a widely accepted convention in RL literature (e.g., in Bellman equations and value function definitions), which allows for concise expressions without explicitly writing the expectation each time.
> > > > > > > >
> > > > > > > > The simplification does not affect the correctness of the reward bound derived using the total variation (TV) distance. In our analysis, the expected cumulative reward difference $\Delta R $ is expressed as:
> > > > > > > >
> > > > > > > > $$
> > > > > > > > \Delta R = \frac{\lambda}{1 - \gamma} \sum_{s, a} (d^\mu(s, a) - d^\pi(s, a)) R(s, a),
> > > > > > > > $$
> > > > > > > >
> > > > > > > > where $ R(s, a) = \mathbb{E}\_{s' \sim T(\cdot | s, a)} [R(s, a, s')] $. Since $R(s, a) $ is a bounded function (with $|R(s, a)| \leq R_{\max} $), the standard TV bound applies:
> > > > > > > >
> > > > > > > > $$
> > > > > > > > \left| \sum_{s, a} (d^\mu(s, a) - d^\pi(s, a)) R(s, a) \right| \leq 2 \cdot \mathrm{TV}(d^\mu, d^\pi) \cdot R_{\max},
> > > > > > > > $$
> > > > > > > >
> > > > > > > > Leading to:
> > > > > > > >
> > > > > > > > $$
> > > > > > > > |\Delta R| \leq \frac{2 \lambda R_{\max}}{1 - \gamma} \cdot \mathrm{TV}(d^\mu, d^\pi).
> > > > > > > > $$
> > > > > > > >
> > > > > > > > This derivation remains valid because $ R(s, a) $ correctly handles the expectation over transitions, ensuring the bound holds as presented. To further clarify, we have made an explicit note in Section 4.1 that $ R(s, a) = \mathbb{E}_{s' \sim T(\cdot | s, a)} [R(s, a, s')] $, bridging the notation between the two sections.
> > > > > > > >
> > > > > > > > We appreciate the reviewer's attention to detail and believe this clarification addresses the concern while confirming the mathematical soundness of our analysis.

---

### Review · Reviewer_Xnhu · 2025-02-25

**Summary Of Contributions:**

This paper introduces Meta Offline-Online Reinforcement Learning (MOORL), a framework designed to integrate offline and online reinforcement learning. The core idea is to train a "meta-policy" that can effectively adapt to the distributional shifts between offline and online data sources. MOORL aims to improve sample efficiency and stability compared to purely online or offline RL methods, and to do so without requiring extensive, task-specific design choices or hyperparameters. The authors provide a theoretical analysis on the benefits of mixing offline/online data as a regularization, and empirically evaluate MOORL on a variety of benchmark tasks (D4RL and V-D4RL), including both state-based and pixel-based environments.

**Audience:**

Yes

**Claims And Evidence:**

Yes

**Requested Changes:**

- Could the authors provide details about the number of training/environment steps for each task as well as learning curves if possible?
- I think it is important to have the comparison between ReBRAC, RLPD and MOORL in the main text of the paper. It is a core claim of the paper that MOORL is design free and the evidence to support this should be in the main text and not the appendix.
- The authors make a note about the increased computational overhead of running RLPD however there aren't any quantitative numbers for this claim, such as training time, gradient steps, memory usage, etc. It isn't necessary to provide all of these, but at least some quantitative metrics would be useful.
- Could the authors compare performance on versions of Adroit with suboptimal/random data mixed in? For example with 20% and 50% of random data mixed with the expert human demonstrations? It would strengthen their claim that "MOORL performs most optimally across tasks, specifically where offline data is suboptimal." Especially since there isn't a clear leader in performance in this task.
- Could the authors provide the standard deviation of the Q Values for the experiment in Fig 1?
- Are there any meta RL baselines that are relevant for comparison?
- Since the algorithm randomly chooses at every epoch which buffer to perform a fixed number of inner gradient steps, wouldn't this be a problem when the buffers are of significantly different sizes? Wouldn't this bias the algorithm ?
- The hyperparams given in Appendix A are for MOORL and not RLPD I assume? Could you provide the hyper parameters for the RLPD/other baselines and if they needed per task tuning?

**Strengths And Weaknesses:**

Strengths
- Empirical Evaluation: The experimental results are comprehensive, covering a wide range of tasks, including state and pixel based observation spaces
- Relevant Problem: The effective integration of offline data to jumpstart the RL process is an open area of research and is highly important for the greater adoption of RL.
- MOORL appears to be robust to task changes and doesn't need any task specific design choices. It is preferable to have algorithms that can be applied to new problems without needing extensive tuning.


Weaknesses
- Although the paper claims to be "design-free," meta-learning itself introduces hyperparameters (e.g., meta, n, K in Algorithm 1). The paper doesn't discuss the sensitivity of MOORL to these hyperparameters.
- The paper makes repeated claims about RLPD that is "requires tailored design parameters and careful task-specific tuning". However they do not say that they needed extensive tuning of RLPD for the different tasks and (in the absence of contrary evidence) appear to have used a single set of hyperparam choices for all tasks.
- See questions below

---

> ### Author Response · Authors · 2025-03-06
> **Response to Reviewer Xnhu**
>
> We appreciate the reviewer's efforts and insights to improve the work's quality. We have incorporated the suggestions you made in the revised draft. Below, we address specific points you raised.
>
> **Requested changes 1: could the authors provide details about the number of training/environment steps for each task as well as learning curves, if possible?**
>
> We train policy for $300K$ timesteps across tasks with $4$ inner distribution adaption gradient steps and $1$ meta-update at each time step, resulting in a total of $1.5M$ gradient steps. We have also added training curves in the revised draft. can be found in the Appendix Figure 2.
>
> **Requested changes 2:   I think it is important to have the comparison between ReBRAC, RLPD and MOORL in the main text of the paper**
>
> We will move the comparison section to the main text in the final draft of the paper.
>
> **Requested changes 3: The authors make a note about the increased computational overhead of running RLPD however there aren't any quantitative numbers for this claim.**
>
> We have included the quantitative numbers to highlight the computational efficiency of MOORL (Appendix D). Specifically, when the policy is trained for 300K time steps, RLPD with UTD 20 performs 6M gradient steps while MOORL performs similarly using only 1.5M  gradient steps while being large ensemble-free. The large Q-ensemble and high UTD causes RLPD to take around 0.50 sec per timestep on a single RTX A4000 GPU, while MOORL takes 0.05 sec per timestep.
>
> **Requested changes 4: Could the authors compare performance on versions of Adroit with suboptimal/random data mixed in? For example with 20\% and 50\% of random data mixed with the expert human demonstrations?**
>
> We appreciate the reviewer’s suggestion to evaluate MOORL on Adroit with mixed suboptimal/random data. While we do not currently have results for these specific settings, our findings on the `locomotion-random' datasets suggest that MOORL benefits from diverse, low-return data by leveraging online adaptation. However, we acknowledge that explicitly evaluating different proportions of mixed suboptimal data would provide a more direct validation of MOORL’s robustness in such scenarios. We will discuss this limitation in the paper and highlight it as a valuable direction for future investigation.
>
> **Requested changes 5: Could the authors provide the standard deviation of the Q Values for the experiment in Fig 1?**
>
> We have added the standard deviation of the Q Values for the experiment in Fig 1.
>
> **Requested changes 6: Are there any meta RL baselines that are relevant for comparison?**
>
> Ours is the first approach that uses meta-learning to integrate offline data with online learning. There are no other baselines that use meta-learning in the hybrid RL setting.
>
> **Requested changes 7: when the buffers are of significantly different sizes? Wouldn't this bias the algorithm ?**
>
> The primary motivation for incorporating offline data is to expose the agent to diverse scenarios, thereby improving exploration and generalization. However, if the offline dataset lacks sufficient diversity or coverage, it may introduce biases that negatively impact policy learning. In particular, when the offline data is highly skewed or unrepresentative of the overall task, MOORL—like other hybrid RL methods (e.g., RLPD)—can experience reduced learning efficiency. To mitigate potential biases, one approach is to use different learning rates for different data distributions or adopt a non-uniform buffer selection strategy. As a future direction, an adaptive curriculum (tzannetos2023proximal) could be designed to dynamically adjust buffer selection, ensuring a balanced learning process across offline and online data.
>
> **Requested changes 8 and Weakedness 2: Could you provide the hyperparameters for the RLPD/other baselines and if they are needed per task tuning?**
>
> The hyperparameters provided are for MOORL, which are the same for RLPD. However, RLPD introduces many other task-specific design elements as well, as described in Appendix Table $7$. For example, RLPD uses a deeper network for an antmaze task and different UTD for state-based and pixel-based tasks. For Antmaze and locomotion, RLPD uses CDQ; for locomotion, RLPD uses entropy backup. Further, RLPD uses many other design elements like layer norm, high UTD, and large Q-ensemble.
>
> **Weakenss 1: The paper doesn't discuss the sensitivity of MOORL to these hyperparameters ($K$)**
>
> We have added an ablation study highlighting the effect of the number of inner gradient steps $K$. Through the ablation study, we found that for a small inner gradient step, the agent shows slow learning as a generalization to changing distribution becomes difficult, while for the higher values, it provides no added advantages. A more detailed discussion can be found in Appendix A.
>
> [1] Tzannetos, Georgios, et al. "Proximal curriculum for reinforcement learning agents." arXiv preprint
> arXiv:2304.12877 (2023).

---

### Review · Reviewer_xTLd · 2025-02-28

**Summary Of Contributions:**

The paper introduces MOORL, a meta-learning framework that unifies offline and online reinforcement learning. It formulates policy adaptation as a bi-level optimization, leveraging KL-regularized performance bounds to balance exploration and exploitation. MOORL uses a Reptile-style meta-update to integrate offline pre-training with online fine-tuning without additional hyperparameters. Theoretical analysis reveals how mixing offline and online data influences policy improvement. Empirical results across 28 tasks demonstrate superior stability and sample efficiency over prior methods. Unlike existing approaches, MOORL automates the balance between data sources, providing a general and principled solution for hybrid reinforcement learning.

**Audience:**

Yes

**Claims And Evidence:**

Yes

**Requested Changes:**

1. Discuss how pathological offline data affects MOORL. If offline data comes from a poor policy, does MOORL still help, or can it mislead learning? The success on “random” datasets suggests that diverse but low-return data aids exploration. However, if offline data is highly biased, could it restrict exploration due to KL penalties? Consider whether MOORL avoids this issue by alternating data sources instead of mixing them directly. Conclude that MOORL may naturally shift to online data if offline data is suboptimal, highlighting this as a limitation or future work discussion point.
2. Include Ablation Studies (Meta-Parameters): Since MOORL introduces the meta-learning procedure, an ablation or sensitivity analysis on its hyperparameters would be very insightful. Two suggestions: (a) Vary the number of inner-loop gradient steps N (e.g., try $N=1$ vs $N=4$ vs $N=8$) to see how it affects performance. And discuss the relationship of UTD from RLPD.

**Strengths And Weaknesses:**

**Strengths**

1. Novel integration of meta-learning with hybrid RL, enabling dynamic adaptation to distribution shifts.
2. Comprehensive experiments across diverse tasks (locomotion, navigation, manipulation, pixel-based).
3. Strong theoretical grounding in analyzing performance bounds and KL divergence effects.

**Weaknesses**

1. Computational costs of meta-learning (e.g., inner-loop updates) are not quantified. The meta-learning approach introduce extra computation burden.

---

> ### Author Response · Authors · 2025-03-06
> **Response to  Reviewer xTLd**
>
> We appreciate the reviewer’s insightful suggestion. Below, we address all the concerns and suggestion:
>
> **Request changes 1: Effect of pathological offline data**
>
> Our results on "random" datasets suggest that diverse, low-return offline data can still facilitate exploration, likely due to its broad coverage. However, if offline data is highly biased—collected from a narrow, suboptimal policy—MOORL may face challenges due to the KL regularization. This could, in principle, restrict exploration in the online phase if the offline data is heavily biased and lacking in diversity. However, unlike conventional hybrid RL methods that directly mix offline and online data, MOORL alternates between these sources and meta-learns a policy that adapts separately to each distribution. This design helps prevent direct bias transfer, as the inner-loop fine-tuning step ensures that the policy is not overly constrained by offline data during online adaptation. As a result, when offline data is suboptimal, MOORL naturally shifts toward relying more on online interactions. That said, if the offline dataset is both biased and narrow, the imposed KL constraints might still limit exploration. As a potential future direction, incorporating mechanisms to dynamically evaluate the quality of offline data and down-weight misleading samples could enhance learning stability and robustness. Additionally, a curriculum-based strategy (tzannetos2023proximal) that adaptively selects offline and online data could further improve robustness in highly biased settings. We have included these insights in the revised draft and highlighted them as future work direction.
>
> **Request changes 2: Ablation study of parameter $K$ and its relation with UTD**
>
> We have added an ablation study highlighting the effect of the number of inner gradient steps $K$. Through the ablation study, we found that for a small inner gradient step, the agent shows slow learning as a generalization to changing distribution becomes difficult, while for the higher values, it provides no added advantages. A more detailed discussion can be found in Appendix A.
>
> We have included the distinction of UTD in RLPD with the number of inner gradient steps in the MOORL, which can be found in Appendix C. Though inner gradient steps seem to have some similarities with UTD in RLPD, there is a subtle difference. The UTD dictates the number of gradient updates per environment step, primarily influencing sample efficiency. On the other hand, our adaptation steps ($K$) govern the number of inner-loop within each outer-loop meta-update
> iteration, directly shaping the adaptation dynamics rather than the frequency of gradient updates.
>
> **Weakness 1: Computation cost comparison**
>
> We have included details on computational cost comparison, which can be found in Appendix D. Specifically, when the policy is trained for 300K time steps, RLPD with UTD 20 performs 6M gradient
> steps while MOORL performs similarly using only 1.5M gradient steps while being large ensemble-free. The
> large Q-ensemble and high UTD causes RLPD to take around 0.50 sec per timestep on a single RTX A4000
> GPU, while MOORL takes 0.05 sec per timestep.
>
> [1] Tzannetos, Georgios, et al. "Proximal curriculum for reinforcement learning agents." arXiv preprint arXiv:2304.12877 (2023).

---

### Decision · Action_Editor_r8jM · 2025-04-28

**Recommendation:** Accept with minor revision

**Comment:**

The paper shows an interesting perspective on combining offline and online training in RL. After extensive adjustments based on the reviews and the discussion and the resulting improvements, the paper can be accepted.

However, a revision is still necessary. The paper still needs to be proofread very carefully. Here is a list, which is not exhaustive:

* Question marks must be resolved, e.g. “(?Nair et al. ”and “(?)”
* “Antmaze” -> ‘AntMaze’

* In the bibliography, the entry for Adam is incorrect: “D Kinga, Jimmy Ba Adam, et al. A method” -> "D Kinga, Jimmy Ba, Adam:. A method“
”California;, 2015." -> “California, 2015.”

* In many places letters are incorrectly written in lower case, “bellman”, “rl”, ‘ppo’, “q”

* In Tables 1 to 4, the $\pm$ sign is used together with the standard deviation. The $\pm$ sign should exclusively be used to express an  uncertainty, e.g. the 95% confidence interval, or the standard deviation of the mean (also called standard error). I suggest to express the standard error instead of the standard deviation by dividing the standard deviation by sqrt(10 - 1) = 3. E.g. “93.4$\pm$ 0.2” instead of “93.4 $\pm$ 0.4” (uncertainties are always rounded up and given with one or two valid digits, the number of decimal places of the measured value (i.e. the number before the $\pm$ sign) must correspond to the number of decimal places of the uncertainty).
If the authors insist on specifying the standard deviation instead of an uncertainty, then brackets should be used instead of the $\pm$ sign, for example, or a separate column should be used.

**Audience:**

The reviewers agree that the paper is interesting. In particular, it is emphasized that it shows an interesting perspective on combining offline and online training in RL.

**Claims And Evidence:**

After extensive adjustments, especially in the discussion with Reviewer uLsg and the resulting improvements, the claims are considered to be sufficiently supported.

---

> ### Author Response · Authors · 2025-05-13
>
> Dear AE,
>
> We sincerely thank the reviewers and the action editor for their thoughtful and constructive feedback, which greatly improved the clarity and quality of our manuscript. We will make further changes and post the camera-ready version soon.
>
> We have uploaded the camera-ready version of the manuscript. We have proofread the manuscript and incorporated all the changes suggested by you, including writing issues, corrected bibliographical errors, and revised tables highlighting the standard deviation of the mean.
>
> Please let us know if there are further issues.
>
> Authors 4150

---

> ### Comment · Action_Editor_r8jM · 2025-05-13
> **A few more corrections**
>
> Dear authors,
>
> I  welcome the fact that the results are now given with uncertainty. However, care must still be taken to ensure that the number of decimal places of the uncertainty and the measured value match. This is not always the case in tables 1, 2, 3, and 4.
>
> Table 1
>
> 93.4 $\pm$ 0.13 -> 93.4 $\pm$ 0.2 (uncertainties are always rounded up) or 93.4x $\pm$ 0.13 (according to the rule, two decimal places are only used for the uncertainty if it is less than 0.15 (or 1.5). I do not know the value of the decimal place x.
>
> 89.1 $\pm$ 1.87 -> 89 $\pm$ 2 (89.1 $\pm$ 1.9 is also acceptable)
>
> 45.0 $\pm$ 0.37 -> 45.0 $\pm$ 0.4
>
> 54.7 $\pm$ 0.30 -> 54.7 $\pm$ 0.3
>
> 30.9 $\pm$ 0.13 -> 30.9 $\pm$ 0.2 (uncertainties are always rounded up) or 30.9x $\pm$ 0.13 (according to the rule, two decimal places are only used for the uncertainty if it is less than 0.15 (or 1.5). Whereby I do not know the value of the decimal place x.
>
> And so on, also in tables 2, 3 and 4.
>
> Furthermore, the correct use of spaces must be checked, e.g. 98.5$\pm$ 0.40 -> 98.5 $ \pm$ 0.4
>
> What is the intention of the leading zeros, at 02.0 or 00.0? I assume that the leading zeros should be omitted.
>
> The entry in the bibliography for Adam is still incorrect. There are two authors “Diederik P. Kingma and Jimmy Ba” and no “et al.”
>
> There are still unusual and incorrect lower case letters in the bibliography, „bellman“, „q-learning

---

> > ### Author Response · Authors · 2025-05-21
> >
> > We apologize for the unintended oversight in the camera-ready version. We have uploaded the revised manuscript and addressed all the points raised by you.